# ReWatch-R1: Boosting Complex Video Reasoning in Large Vision-Language Models through Agentic Data Synthesis

**Congzhi Zhang**[*1,2,3]     **Zhibin Wang**[*1,2]     **Yinchao Ma**[*1,2,4]     **Jiawei Peng**[3]

**Yihan Wang**[5]   **Qiang Zhou**[1,2]   **Jun Song**[†1,2]   **Bo Zheng**[1,2]

[1]Alibaba Group Holding Limited    [2]Future Living Lab of Alibaba    [3]SEU    [4]USTC    [5]DUT

*zhangcongzhi0@gmail.com*    *jsong.sj@alibaba-inc.com*

## Abstract

While Reinforcement Learning with Verifiable Reward (RLVR) significantly advances image reasoning in Large Vision-Language Models (LVLMs), its application to complex video reasoning remains underdeveloped. This gap stems primarily from a critical data bottleneck: existing datasets lack the challenging, multi-hop questions and high-quality, video-grounded Chain-of-Thought (CoT) data necessary to effectively bootstrap RLVR. To address this, we introduce *ReWatch*, a large-scale dataset built to foster advanced video reasoning. We propose a novel multi-stage synthesis pipeline to synthesize its three components: *ReWatch-Caption*, *ReWatch-QA*, and *ReWatch-CoT*. A core innovation is our **Multi-Agent ReAct framework** for CoT synthesis, which simulates a human-like "re-watching" process to generate video-grounded reasoning traces by explicitly modeling information retrieval and verification. Building on this dataset, we develop **ReWatch-R1** by post-training a strong baseline LVLM with Supervised Fine-Tuning (SFT) and our RLVR framework. This framework incorporates a novel **Observation & Reasoning (O&R) reward mechanism** that evaluates both the final answer's correctness and the reasoning's alignment with video content, directly penalizing hallucination. Our experiments show that ReWatch-R1 achieves **state-of-the-art performance** on five challenging video reasoning benchmarks. Project Page.

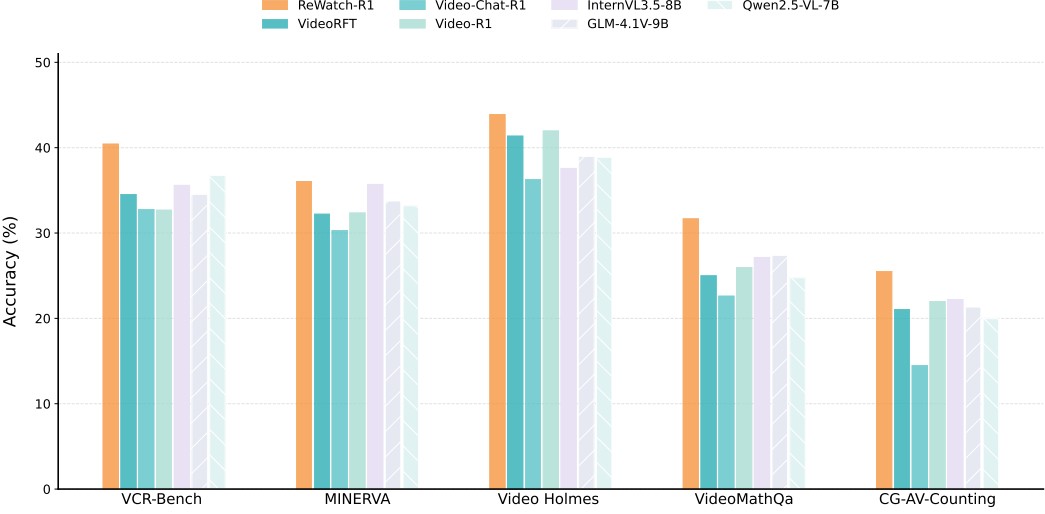

Figure 1: **Performance comparison of our ReWatch-R1 with previous state-of-the-art LVLMs on five video reasoning benchmarks.** Except for Qwen2.5-VL-7B, all other models use thinking mode. All models were evaluated at 192 frames.

---

*Equal contribution.

†Corresponding Author.

# 1 INTRODUCTION

While the training paradigm of Supervised Fine-Tuning (SFT) combined with Reinforcement Learning with Verifiable Reward (RLVR) [18; 42] significantly advances image reasoning in Large Vision-Language Models (LVLMs) [52; 22; 51], its application to complex video reasoning remains nascent. Recent open-source video models [16; 30; 45; 10; 39] trained with SFT+RLVR still underperform on high-difficulty benchmarks, especially for multi-step temporal tasks such as causality, state tracking, and counting events across long videos [40; 38; 11; 41; 36].

Recent efforts to apply the SFT+RLVR paradigm to video [16; 30; 45; 10; 39] typically bootstrap the SFT phase with CoT data synthesized from existing simple video QA datasets, before applying RLVR. However, this approach is fundamentally undermined by the quality of the underlying data. As illustrated in Figure 2(left), prevailing open-source data [16] suffers from three flaws: (1) **holistic, untimestamped captions** that erase temporal structure; (2) **simple, perception-based QA** that can be answered from short clips or textual priors; and (3) **visually unfaithful CoT** that relies on commonsense knowledge and process of elimination. This data bottleneck prevents SFT from teaching true video-grounded reasoning, and the subsequent RL phase, lacking a reliable reward signal for process correctness, struggles to penalize hallucination and improve logical fidelity [12; 23].

To address these limitations, we introduce **ReWatch**, a large-scale dataset explicitly designed to foster advanced video reasoning. **ReWatch** is constructed through a multi-stage synthesis pipeline and comprises three tightly coupled components: *ReWatch-Caption*, *ReWatch-QA*, and *ReWatch-CoT*. First, **ReWatch-Caption provides temporally dense video descriptions**. We employ a hierarchical captioning method to generate detailed, timestamped narratives that form a high-fidelity foundation for complex reasoning. Second, **ReWatch-QA features high-difficulty question-answer pairs**. We use a contrastive generation strategy, creating questions from detailed captions that cannot be answered by concise summaries, and apply a three-tier filter to guarantee video dependency. Finally, **ReWatch-CoT promotes video-grounded reasoning**. We employ a novel Multi-Agent ReAct framework to synthesize CoT that simulates a human-like "re-watching" process. This generates reasoning traces that explicitly document information retrieval and verification against the video content. As shown in Figure 2(right), our *ReWatch* data delivers **high-fidelity captions, high-difficulty QAs, and video-grounded CoTs**.

Building on *ReWatch*, we post-train a strong LVLM in two stages to obtain **ReWatch-R1**. After an initial SFT phase that teaches step-by-step reasoning, we employ RLVR augmented with a novel

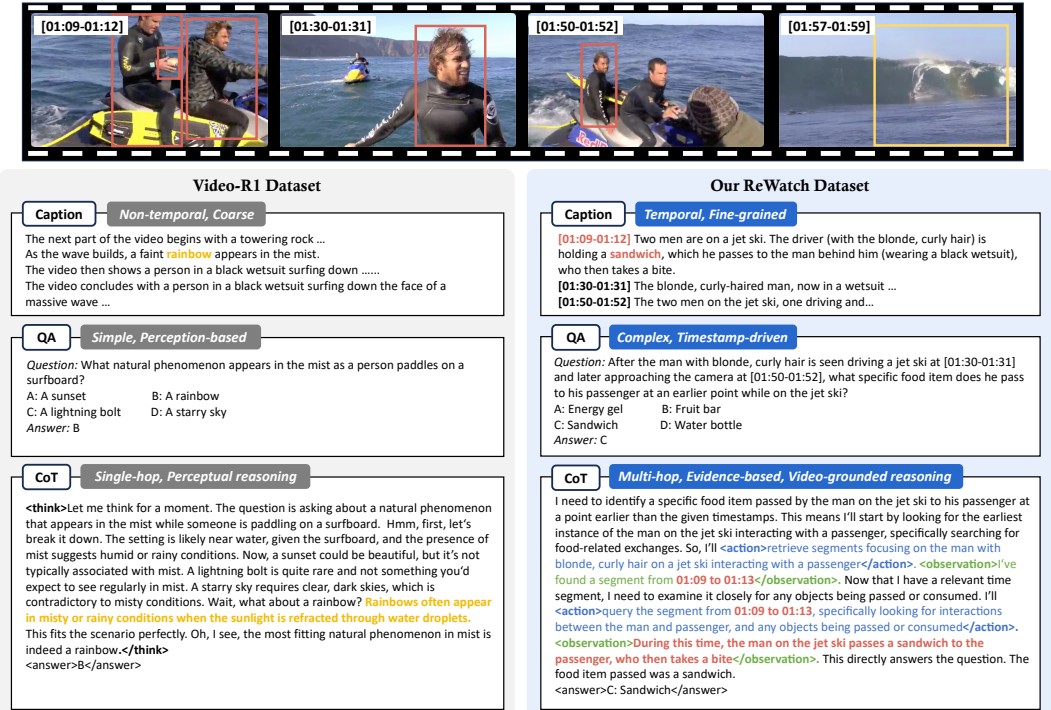

Figure 2: **A comparative of *ReWatch* dataset and *Video-R1* dataset on the same source video.**

**Observation & Reasoning (O&R) reward**. Unlike rewards that score only the final answer, O&R also evaluates whether intermediate observations are factually supported by the video and whether the reasoning is sufficient to recover the correct answer from those observations. This dual emphasis on process and outcome explicitly incentivizes verifiable, evidence-linked reasoning, reducing hallucinations and improving logical consistency. As summarized in Figure 1, **ReWatch-R1 sets new state of the art on five challenging video reasoning benchmarks**, substantially outperforming models trained on alternative open-source data.

In summary, our contributions are:

- A novel, multi-stage agentic pipeline for synthesizing a large-scale, high-quality video reasoning dataset (*ReWatch*).
- A new Observation & Reasoning (O&R) reward for RLVR that improves reasoning by rewarding both final-answer correctness and the factual grounding of intermediate steps in video content.
- ReWatch-R1, a post-trained LVLM that achieves state-of-the-art results on five complex video reasoning benchmarks.

## 2 DATA CONSTRUCTION: THE REWATCH DATASET

To address the above data bottlenecks, we introduce ***ReWatch***, a large, high-fidelity, high-difficulty, and video-grounded dataset for advanced video reasoning. As shown in Figure 3, it is constructed in three stages: **Hierarchical Video Captioning**, **High-Difficulty QA Generation**, and **Multi-Agent CoT Synthesis**. The dataset contains 10k captions, 170k QA pairs, and 135k CoTs. More details and statistics are in Appendix B.

### 2.1 STAGE 1: HIERARCHICAL VIDEO CAPTIONING

To address the hallucination issue in LVLMs when processing long videos and to generate high-fidelity video descriptions, we propose a **Hierarchical Dynamic Frame-Rate Generation** pipeline for our *ReWatch-Caption-10k* dataset. The process is applied to our video corpus $\mathcal{V}$, sourced from five public datasets [24; 19; 32; 16; 59].

**Semantic Segmentation.** For each video $V \in \mathcal{V}$, we first partition $V$ into $k$ semantically coherent segments $S$ using LVLM $\mathcal{M}_{\text{seg}}$, at a low-frame-rate. To strictly preserve long-term contextual integrity, we apply this segmentation only to videos exceeding 10 minutes in duration. Unlike fixed-interval splitting, our approach leverages the LVLM to perform semantic-based partitioning, ensuring that each segment $s_i$ retains a complete narrative structure with an approximate duration of 10 minutes. Each segment $s_i$ corresponds to a temporal interval $[t_i^{\text{start}}, t_i^{\text{end}}]$, preserving event integrity.

$$S = \{s_1, \ldots, s_k\} = \mathcal{M}_{\text{seg}}(V) \tag{1}$$

**Detailed Description Generation.** We use a powerful LVLM $\mathcal{M}_{\text{cap}}$ to process each segment $s_i$ at a high frame rate and generate a detailed description $D_i^{\text{rel}}$, which includes $m_i$ distinct events $\{c_{ij}\}$ along with their relative timestamps $\{\tau_{ij}\}$.

$$D_i^{\text{rel}} = \{(c_{ij}, \tau_{ij})\}_{j=1}^{m_i} = \mathcal{M}_{\text{cap}}(s_i) \tag{2}$$

**Timestamp Realignment.** Finally, a function $\mathcal{P}$ converts relative timestamps $\tau_{ij}$ to absolute ones $t_{ij}$ by adding the segment's start time.

$$t_{ij} = \mathcal{P}(\tau_{ij}, t_i^{\text{start}}) = t_i^{\text{start}} + \tau_{ij} \tag{3}$$

The final video caption $C_{\text{detail}}(V)$ is the union of all timestamped descriptions.

$$C_{\text{detail}}(V) = \bigcup_{i=1}^{k} \{(c_{ij}, t_{ij})\}_{j=1}^{m_i} \tag{4}$$

This hierarchical approach generates temporally precise and semantically rich descriptions while avoiding the hallucination issues associated with LVLMs processing long videos.

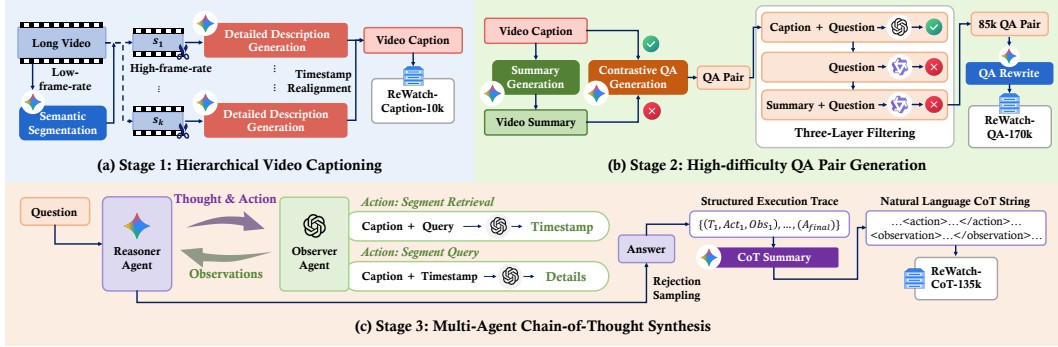

Figure 3: **The data construction pipeline. (a) Caption Construction.** Long videos are semantically segmented to produce detailed, temporally-aware captions. **(b) QA Pair Generation.** A contrastive method using detailed and summary captions generates complex questions, which are then purified by a three-layer filtering mechanism. **(c) CoT Synthesis.** A ReAct framework with a Reasoner Agent and an Observer Agent simulates a "re-watching" process by performing targeted queries on the video caption to generate video-grounded reasoning traces.

## 2.2 STAGE 2: HIGH-DIFFICULTY QA PAIR GENERATION

To create our *ReWatch-QA-170k* dataset, we design a pipeline to generate challenging QA pairs requiring fine-grained video analysis. It combines **Contrastive Prompting** with **Three-Layer Filtering**.

**Contrastive QA Generation.** Given a detailed caption $C_{\text{detail}}$, we first generate a concise summary $C_{\text{sum}} = \mathcal{M}_{\text{sum}}(C_{\text{detail}})$ using a lightweight LLM. Then, inspired by previous work [73; 5], our QA generator $\mathcal{M}_{\text{qa}}$ processes both $C_{\text{detail}}$ and $C_{\text{sum}}$ to create QA pairs $(Q, A)$ that are explicitly answerable from the detailed caption but not from the summary alone. This ensures questions probe fine-grained details while excluding trivial ones.

$$(Q, A)_{\text{raw}} = \mathcal{M}_{\text{qa}}(C_{\text{detail}}, C_{\text{sum}}) \tag{5}$$

To guide generation and ensure diversity, we pre-define 10 question types.

**Three-Layer Filtering.** Raw pairs undergo a three-layer filtering cascade to ensure quality and video-dependency:

- **Filter 1: Answer Verification, $\mathcal{F}_1$:** A verifier $\mathcal{M}_{\text{verify}}$ confirms the factual correctness of the answer based on $C_{\text{detail}}$.

$$(Q, A) \text{ passes } \mathcal{F}_1 \iff \mathcal{M}_{\text{verify}}(Q, A, C_{\text{detail}}) = \text{True} \tag{6}$$

- **Filter 2: Text Bias Elimination, $\mathcal{F}_2$:** Ensures the question is unanswerable from general knowledge by probing a set of LLMs $\mathbb{M}_{\text{probe}}$.

$$(Q, A) \text{ passes } \mathcal{F}_2 \iff \frac{1}{|\mathbb{M}_{\text{probe}}|} \sum_{\mathcal{M} \in \mathbb{M}_{\text{probe}}} \mathbf{1}(\mathcal{M}(Q) \approx A) < \theta_{\text{text}} \tag{7}$$

- **Filter 3: Summary Bias Elimination, $\mathcal{F}_3$:** Similarly ensures the question is unanswerable using the summary $C_{\text{sum}}$.

$$(Q, A) \text{ passes } \mathcal{F}_3 \iff \frac{1}{|\mathbb{M}_{\text{probe}}|} \sum_{\mathcal{M} \in \mathbb{M}_{\text{probe}}} \mathbf{1}(\mathcal{M}(Q, C_{\text{sum}}) \approx A) < \theta_{\text{sum}} \tag{8}$$

Where $\theta_{\text{text}}$ and $\theta_{\text{sum}}$ are threshold for consensus. The 85k pairs passing all filters are then rewritten by LLM $\mathcal{M}_{\text{rewrite}}$ into multiple-choice questions, yielding a total of 170k QA pairs.

### 2.3 STAGE 3: MULTI-AGENT CHAIN-OF-THOUGHT SYNTHESIS

To generate our *ReWatch-CoT-135k* dataset, we introduce a multi-agent ReAct-based framework that explicitly construct the video-grounded CoT. This method externalizes the observation process for active information retrieval.

We define two agents: a **Reasoner** $\mathcal{A}_R$ that produces thoughts $T$ and actions $Act$, and an **Observer** $\mathcal{A}_O$ that executes actions on the video caption $C_{\text{detail}}$ to return observations $Obs$.

For a given question $Q$, the agents interact in a loop. At each step $t$, the Reasoner uses the history $H_{t-1} = (Q, T_1, Act_1, Obs_1, \ldots, T_{t-1}, Act_{t-1}, Obs_{t-1})$ to decide the next step:

$$(T_t, Act_t) = \mathcal{A}_R(H_{t-1}) \tag{9}$$

The Observer executes the action to retrieve information from the video context:

$$Obs_t = \mathcal{A}_O(Act_t, C_{\text{detail}}) \tag{10}$$

It is important to clarify that our Observer Agent $\mathcal{A}_O$ retrieves observations from the detailed textual captions ($C_{detail}$) rather than processing raw video frames during synthesis. Through manual inspection, we confirmed that our hierarchical captions from Stage 1 are sufficiently fine-grained to serve as a high-fidelity proxy for visual content. This text-based simulation drastically improves the efficiency and scalability of data synthesis compared to pixel-based methods. While our current pipeline is text-based, the synthesized 'Thought-Action-Observation' trajectories provide a foundational resource for training future 'thinking-with-video' models that can directly query visual encoders.

This process continues until the Reasoner produces a final answer. The core actions $Act_t$ simulate visual lookup:

- `segment_retrieval(query)`: Finds the timestamp of an event from a natural language query.
- `segment_query(timestamp)`: Retrieves the detailed description of an event from a timestamp.

This entire text-based simulation is highly efficient. The structured execution trajectory $\mathcal{T} = \{(T_1, Act_1, Obs_1), \ldots, (A_{\text{final}})\}$ is then converted by LLM $\mathcal{M}_{\text{convert}}$ into a natural language CoT string $\mathcal{R}$ with explicit `<action>` and `<observation>` tags, making it ready for supervised fine-tuning and O&R reward calculation.

## 3 POST-TRAING ON REWATCH DATASET

As shown in Figure 4, we use the SFT+RL paradigm to train Qwen2.5-VL. In the SFT stage, we use multi-task objectives to train to obtain **ReWatch-R1-SFT**. In the RL stage, based on the GRPO [18] algorithm and a novel O&R reward mechanism we propose, we obtain **ReWatch-R1**.

### 3.1 SUPERVISED FINE-TUNING STAGE

In this stage, we perform multi-task SFT on a base LVLM using our three datasets: *ReWatch-Caption-10k* ($\mathcal{D}_{\text{Cap}}$), *ReWatch-QA-170k* ($\mathcal{D}_{\text{QA}}$), and *ReWatch-CoT-135k* ($\mathcal{D}_{\text{CoT}}$). The goal is to jointly instill three core abilities: foundational video-text alignment, direct question-answering ("non-thinking" mode), and step-by-step reasoning ("thinking" mode). Crucially, we train the model to switch between these response modes using distinct instruction prompts. For detailed prompt setting during SFT, please refer to Appendix E.2.

The SFT objective is to minimize a composite loss function, $\mathcal{L}_{\text{SFT}}$, which is the sum of the losses from these three tasks. Let the LVLM be denoted by a policy $\pi_\theta$ with parameters $\theta$. The total loss is defined as:

$$\mathcal{L}_{\text{SFT}}(\theta) = \mathcal{L}_{\text{Cap}} + \mathcal{L}_{\text{QA}} + \mathcal{L}_{\text{CoT}} \tag{11}$$

where each component corresponds to a specific learning objective:

**Video-Text Alignment.** We train the model to generate detailed captions ($C_{\text{detail}}$) from videos ($V$).

$$\mathcal{L}_{\text{Cap}} = -\mathbb{E}_{(V, C_{\text{detail}}) \in \mathcal{D}_{\text{Cap}}}[\log \pi_\theta(C_{\text{detail}}|V)] \tag{12}$$

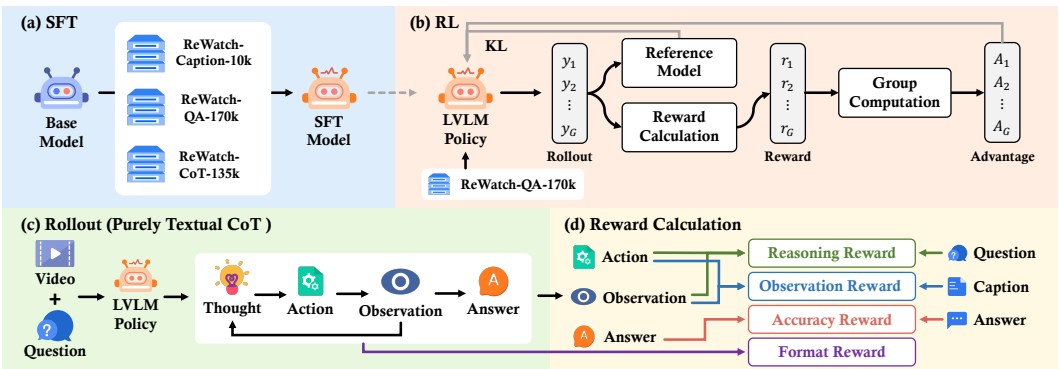

Figure 4: **Our two-stage Post-Training framework.** (a) A Base Model is first fine-tuned (SFT) on all ReWatch datasets, (b) then further refined as a policy via Reinforcement Learning (RL) using the ReWatch-QA dataset. (c) The "Rollout" panel illustrates the generative process of the policy: producing a purely textual chain-of-thought that simulates a Thought-Action-Observation reasoning loop through self-generated text segments. (d) We employ four verifiable reward mechanisms.

**Direct Question-Answering (Non-thinking).** We train the model to output a concise answer ($A$) when given a direct-answer instruction $I_{\text{direct}}$.

$$\mathcal{L}_{\text{QA}} = -\mathbb{E}_{(V,Q,A)\in\mathcal{D}_{\text{QA}}}[\log \pi_\theta(A|V, I_{\text{direct}}, Q)] \tag{13}$$

**Chain-of-Thought Reasoning (Thinking).** We train the model to generate the full reasoning trace ($\mathcal{R}$) when given a think-step-by-step instruction $I_{\text{think}}$.

$$\mathcal{L}_{\text{CoT}} = -\mathbb{E}_{(V,Q,\mathcal{R})\in\mathcal{D}_{\text{CoT}}}[\log \pi_\theta(\mathcal{R}|V, I_{\text{think}}, Q)] \tag{14}$$

By optimizing these objectives concurrently, we produce a versatile SFT Model that is proficient in both direct answering and complex reasoning. This model then serves as the proficient initial policy for the subsequent Reinforcement Learning stage.

## 3.2 REINFORCEMENT LEARNING STAGE

Previous LVLMs of video reasoning [10; 16] directly utilize the accuracy of the final answer $r_{acc}$ as the reward signal for reasoning enhancement through reinforcement learning. Formally,

$$r_{acc} = \mathcal{M}_{judge}(A, A_{gt}), \tag{15}$$

where $\mathcal{M}_{judge}(\cdot)$ is the judge model used to assess the consistency of inputs, which can be a rule-based verifier or an LLM. However, the foundation of video reasoning lies in the ability to reason **grounded in video content**. Such reward for mere accuracy overlooks the capabilities of video content-oriented reasoning, which may lead to potential visual or linguistic hallucinations. To address this limitation, we design the **Observation & Reasoning (O&R) reward mechanism**, which encourages the model to perform appropriate reasoning grounded in the accurate understanding of video content, rather than relying on potential visual or linguistic hallucinations. Specifically, we model the video reasoning QA process as a sequential flow:

$$\text{Video+Question} \rightarrow \text{Observations+Reasoning} \rightarrow \text{Answer}$$

**On one hand**, the model should base its reasoning on accurate observations of the video content. Thus, we first assess the accuracy of video observations in CoT by comparing them with the detailed video caption, and use this evaluation as the observation reward. Formally,

$$\{Act_i, Obs_i\}_{i=1}^{N} = \text{Parse}(\mathcal{R}), \tag{16}$$

$$r_{obs} = \text{mean}(\{\mathcal{M}_{judge}(C_{detail}, \{Act_i, Obs_i\})\}_{i=1}^{N}). \tag{17}$$

Here, $\text{Parse}(\cdot)$ denotes parsing the actions and observations from the model output.

**On the other hand**, the model should reason out appropriate observational actions according to the question. Therefore, we design the reasoning reward by evaluating the accuracy of directly answering

questions using the actions and observations. If the model can provide a correct answer based on these actions and observations, the reasoning process is deemed valid and sufficient. This reward guides the model to reason appropriate observation actions that effectively address the question. Formally,

$$A_{ao} = \mathcal{M}_{infer}(Q, \{Act_i, Obs_i\}_{i=1}^{N}), \tag{18}$$

$$r_{rea} = \mathcal{M}_{judge}(A_{ao}, A_{gt}). \tag{19}$$

Here, $\mathcal{M}_{infer}(\cdot)$ is an LLM used to answer the question based on the given actions and observations. The final reward can be expressed as,

$$r_{O\&R} = r_{acc} \times (1 + r_{obs} + r_{rea}) + r_{fmt}, \tag{20}$$

$$r_{fmt} = \begin{cases} 1, & \text{correct format} \\ 0. & \text{otherwise} \end{cases} \tag{21}$$

Here, $r_{fmt}$ denotes the format reward, enabling the model to output responses in the format we desire. For example, we expect the model to enclose its actions and observations with `<action>...</action>` and `<observation>...</observation>` tags, and the answer with `<answer>...</answer>` tag. Finally, we employ the GRPO [18] algorithm for model optimization.

## 4 EXPERIMENTS

We train Qwen2.5-VL-7B [4] on the *ReWatch* dataset to obtain Rewatch-R1, and then compare it with other LVLMs on five video reasoning and four video understanding benchmarks. For detailed experimental settings, please refer to the Appendix C.1.

### 4.1 MAIN RESULTS

Table 1 shows the superior video reasoning performance of our model, yielding following key insights.

**SOTA Performance among models of a comparable size.** In both 192-frame and 384-frame settings, the average scores of ReWatch-R1 across five reasoning benchmarks significantly surpass those of all other comparison models. This validates the effectiveness of our dataset and training methodology.

**High-Quality CoT Data is Critical.** The SFT-only model ReWatch-R1-SFT (33.25%) already surpasses most competitors like Video-R1-SFT (29.74%) and LongVideoReason-SFT (26.31%), which use the same training configuration. This proves the superiority of our CoT training data.

**RL Unlocks Further Potential.** Reinforcement learning further boosts performance. Our final ReWatch-R1 model improves upon the SFT version (33.25% to 35.51%). This shows that while SFT teaches the form of CoT, our RL phase imparts the spirit, enabling more logical and factually grounded reasoning.

**The Efficacy of "Thinking" is Contingent on Learning "How to Think".** Enabling CoT ("Thinking" mode) is detrimental for an untrained base model (27.54% vs. 30.71%), as it can induce hallucinations. In contrast, our fully trained ReWatch-R1 excels with CoT. This proves our method successfully teaches the model how to reason.

We further evaluate performance on video understanding benchmarks in Table 4 and performance on videos of varying durations in Figure 9. For detailed analysis, please refer to Appendix C.2 and C.3.

### 4.2 ANALYSIS RESULTS

**High-Quality SFT Data is Foundational for RL.** An ablation study in Figure 5a shows two key findings. First, SFT is an indispensable prerequisite for RL, training without it (w/o SFT) causes a catastrophic performance drop, as RL needs a strong initial policy. Second, high-quality CoT data is vital. Replacing our *ReWatch-CoT* data with that from Video-R1 significantly degrades performance. This validates that our multi-agent framework produces a superior training corpus for complex reasoning.

Table 1: **Performance comparison on Video Reasoning tasks.** [*] indicates that we reproduced the model using a training configuration with 192 frames. [†] indicates that reinforcement learning is conducted using exactly the same data as ReWatch-R1. The best results among models of the same size are indicated in **bold**.

| Models | Thinking | VCR Bench | MINERVA | Video Holmes | Video MathQA | CG-AV Counting | Average |
|---|---|---|---|---|---|---|---|
| *192 Frames* | | | | | | | |
| Qwen2.5-VL-32B | ✗ | 39.85 | 38.15 | 43.28 | 33.33 | 23.95 | 35.71 |
| Qwen2.5-VL-7B | ✗ | 36.75 | 33.19 | 38.87 | 24.76 | 19.96 | 30.71 |
| Qwen2.5-VL-7B | ✓ | 34.72 | 29.15 | 34.78 | 24.52 | 14.51 | 27.54 |
| GLM4.1V-9B | ✓ | 34.53 | 33.75 | 38.98 | 27.38 | 21.32 | 31.19 |
| InternVL3.5-8B | ✓ | 30.17 | 33.12 | 35.11 | 27.86 | 22.30 | 29.71 |
| Video-R1 | ✓ | 32.69 | 32.36 | 41.97 | 25.95 | 22.01 | 31.00 |
| Video-Chat-R1 | ✓ | 32.79 | 30.33 | 36.31 | 22.62 | 14.51 | 27.31 |
| VideoRFT | ✓ | 34.53 | 32.22 | 41.37 | 25.00 | 21.03 | 30.83 |
| VersaVid-R1 | ✓ | 36.56 | 31.45 | 39.09 | 24.05 | 23.27 | 30.88 |
| TW-GRPO | ✓ | 26.11 | 34.38 | 42.19 | 26.90 | 19.47 | 29.81 |
| GRPO-CARE | ✓ | 35.49 | 31.87 | 38.27 | 25.48 | 19.57 | 30.14 |
| Video-R1-SFT[*] | ✓ | 33.85 | 31.45 | 37.29 | 26.43 | 19.67 | 29.74 |
| Video-R1-RL[*†] | ✓ | 34.24 | 31.45 | 37.18 | 27.38 | 21.13 | 30.28 |
| LongVideoReason-SFT[*] | ✓ | 24.37 | 29.71 | 38.60 | 23.10 | 15.77 | 26.31 |
| LongVideoReason-RL[*†] | ✓ | 35.30 | 35.01 | 43.49 | 23.57 | 20.55 | 31.58 |
| ReWatch-R1-SFT | ✓ | 35.78 | 35.43 | 39.52 | 30.00 | **25.51** | 33.25 |
| ReWatch-R1 | ✓ | 40.14 | 35.70 | 43.00 | 30.71 | 24.73 | 34.86 |
| + O&R | ✓ | **40.43** | **36.05** | **43.88** | **31.67** | **25.51** | **35.51** |
| *384 Frames* | | | | | | | |
| Qwen2.5-VL-32B | ✗ | 39.75 | 38.63 | 44.04 | 33.81 | 25.71 | 36.39 |
| Qwen2.5-VL-7B | ✗ | 34.91 | 34.59 | 39.90 | 24.76 | 20.16 | 30.86 |
| Qwen2.5-VL-7B | ✓ | 32.45 | 31.10 | 34.89 | 24.00 | 16.57 | 27.80 |
| GLM4.1V-9B | ✓ | 38.59 | 36.54 | 41.10 | **33.10** | 23.08 | 34.48 |
| InternVL3.5-8B | ✓ | 30.56 | 29.43 | 32.55 | 28.57 | 23.27 | 28.88 |
| Video-R1 | ✓ | 32.40 | 35.77 | 41.37 | 23.57 | 20.84 | 30.79 |
| Video-Chat-R1 | ✓ | 31.72 | 31.66 | 36.47 | 22.62 | 14.61 | 27.42 |
| VideoRFT | ✓ | 34.62 | 34.38 | 41.26 | 25.24 | 20.93 | 31.29 |
| VersaVid-R1 | ✓ | 33.46 | 33.75 | 39.74 | 23.57 | 21.32 | 30.37 |
| TW-GRPO | ✓ | 25.82 | 35.43 | 42.24 | 27.86 | 19.96 | 30.26 |
| GRPO-CARE | ✓ | 36.46 | 33.05 | 38.11 | 25.00 | 20.64 | 30.65 |
| Video-R1-SFT[*] | ✓ | 33.95 | 35.56 | 37.29 | 25.24 | 21.91 | 30.79 |
| Video-R1-RL[*†] | ✓ | 35.69 | 32.29 | 37.83 | 26.67 | 20.06 | 30.51 |
| LongVideoReason-SFT[*] | ✓ | 24.18 | 30.20 | 38.49 | 23.33 | 6.04 | 24.45 |
| LongVideoReason-RL[*†] | ✓ | 34.91 | 37.24 | 43.88 | 24.29 | 22.01 | 32.47 |
| ReWatch-R1-SFT | ✓ | 36.17 | 35.50 | 39.09 | 30.48 | 22.78 | 32.80 |
| ReWatch-R1 | ✓ | **39.56** | **38.15** | 43.98 | 30.95 | 25.32 | 35.59 |
| + O&R | ✓ | 38.78 | 36.54 | **44.26** | 32.62 | **26.68** | **35.78** |

**High-quality QA data is crucial for RL.** A comparative analysis in Figure 5b shows that the quality of QA data used for RL determines final performance. Training on only baseline QA data (*Video-R1-QA* [16] (10k) and *LongVideoReason-QA* [10] (10k)) yields the lowest scores (42.0% all, 34.3% reasoning, 51.7% understanding), whereas our *ReWatch-QA* data provides notable improvements. This confirms that *ReWatch-QA*, due to its challenging nature, offers a more potent reward signal that guides the model toward robust reasoning abilities instead of overfitting to simpler patterns.

**Dataset Complexity & Video Dependency.** Figure 6a presents a quantitative analysis of the complexity comparison between the *ReWatch-QA* and *Video-R1-QA* datasets. The detailed experimental design can be found in Appendix C.4. The results show that the *ReWatch-QA* dataset elicits more

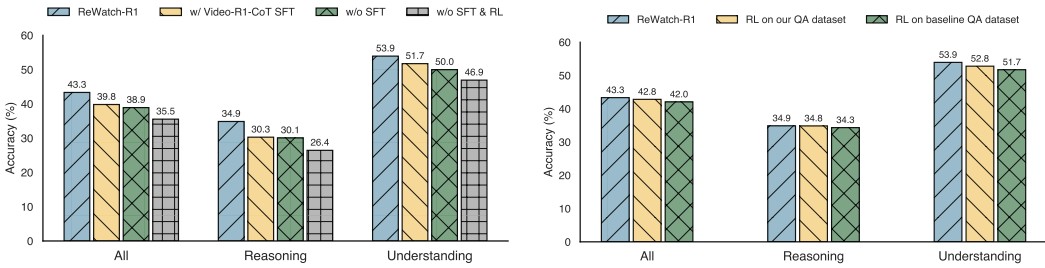

(a) Ablation results using different CoT data.      (b) Ablation results using different QA data.

Figure 5: **Ablation results of our synthesized data against baselines.**

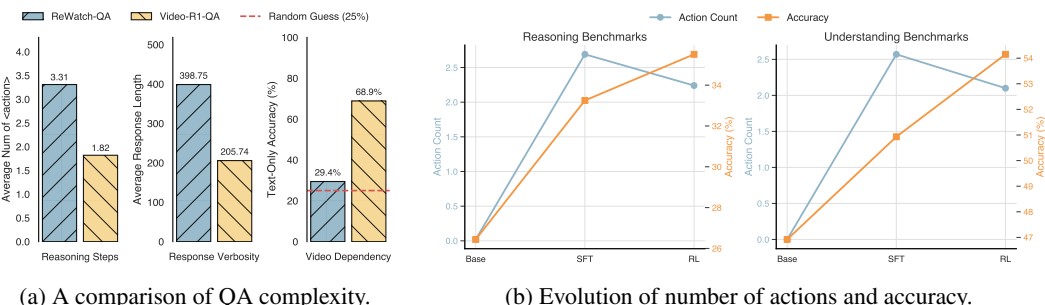

(a) A comparison of QA complexity.      (b) Evolution of number of actions and accuracy.

Figure 6: **Analysis on QA complexity and Evolution of action count.**

profound reasoning than *Video-R1-QA*. *ReWatch* requires nearly double the reasoning steps (3.31 vs. 1.82) and significantly longer responses (398.75 vs. 205.74). Critically, *Video-R1* has a high Text-Only Accuracy of 68.9%, indicating questions are often solvable from text alone. In contrast, the accuracy of ReWatch is only 29.4%, near the 25% random-guess baseline. This proves our three-stage filtering is effective, eliminating textual shortcuts and forcing genuine video understanding.

**RL optimizes the reasoning process, leading to more efficient yet more accurate responses.** Figure 6b shows a two-stage evolution. First, SFT teaches the model a structured reasoning format, increasing action counts and accuracy. Then, during RL, accuracy continues to improve while the average number of actions decreases. This indicates RL refines the policy to be more effective and efficient, pruning redundant steps to focus on critical actions. The model thus transitions from learning reasoning's form (SFT) to mastering its function with efficiency (RL).

**The thinking mode, while converging more slowly during training, ultimately achieves a significantly higher performance ceiling than the non-thinking mode.** As shown in Figure 7, the two modes exhibit different learning dynamics. During the SFT phase (solid lines), the direct-answer "non-thinking" mode improves rapidly, whereas the "thinking" mode develops slowly. This suggests SFT primarily teaches the format of reasoning, not its logic. The subsequent RL phase (dashed lines) acts as a catalyst, causing a dramatic performance leap in the thinking mode by forcing the model to learn the causal links between reasoning and correct answers. Ultimately, the final model's "thinking" performance surpasses the "non-thinking" mode in all tasks. This empirically proves that an explicit, step-by-step reasoning process, cultivated via our SFT-RL regimen, is optimal for complex video tasks.

**Scalability of the ReWatch Framework.** To verify whether our proposed pipeline generalizes to larger parameters, we scaled up the base model to Qwen2.5-VL-32B. As presented in Table 2, the performance trajectory remains consistent with the 7B experiments. The post-trained ReWatch-R1-32B achieves an average accuracy of 38.08% on reasoning benchmarks, surpassing both the strong base model (35.71%) and the SFT variant (36.17%). Notably, the inclusion of the O&R reward mechanism continues to yield performance gains (improving from 37.66% to 38.08%), further validating that our data synthesis pipeline and SFT+RLVR strategy are model-agnostic and effective at unlocking reasoning capabilities in larger-scale LVLMs.

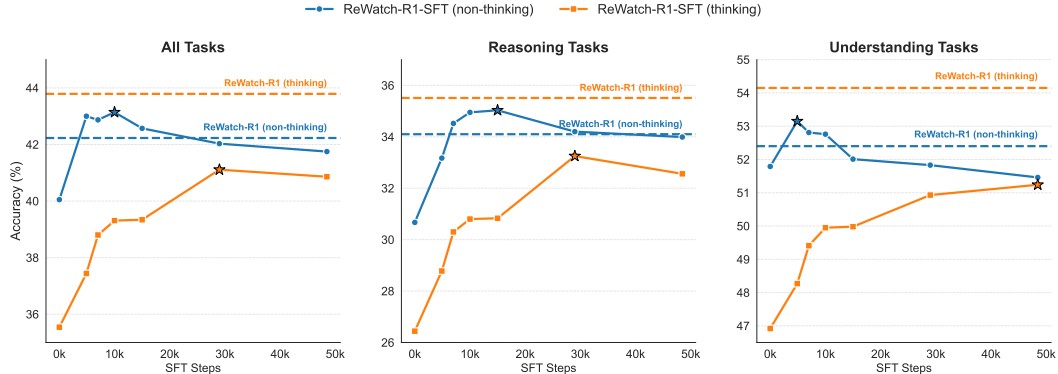

Figure 7: **Impact of SFT and RL on different prompting methods.** The plots show the accuracy of our ReWatch-R1 model with "thinking" (ReAct) vs. "non-thinking" (direct answering) prompting. Solid lines show performance progression during the SFT phase, dashed lines show the final performance after RL.

Table 2: **Performance comparison on Video Reasoning tasks of Qwen2.5-VL-32B.** The best results among models of the same size are indicated in **bold**.

| Models | Thinking | VCR Bench | MINERVA | Video Holmes | Video MathQA | CG-AV Counting | Average |
|---|---|---|---|---|---|---|---|
| Qwen2.5-VL-32B | ✗ | 39.85 | 38.15 | 43.28 | 33.33 | 23.95 | 35.71 |
| ReWatch-R1-SFT-32B | ✓ | 40.81 | 37.52 | 43.11 | 34.29 | **25.12** | 36.17 |
| ReWatch-R1-32B | ✓ | 44.68 | 38.08 | 45.56 | 36.43 | 23.56 | 37.66 |
| + O&R | ✓ | **45.55** | **38.35** | **45.78** | **37.62** | 23.08 | **38.08** |

## 5 CONCLUSION

In this work, we address the critical data bottleneck in complex video reasoning by introducing *ReWatch*, a large-scale dataset synthesized via a novel multi-stage agentic pipeline that generates temporally-dense captions, challenging multi-hop questions, and video-grounded Chain-of-Thought traces. We then develop ReWatch-R1 by post-training a strong LVLM using an SFT and RLVR framework, featuring our innovative Observation & Reasoning (O&R) reward that uniquely evaluates both the correctness of the final answer and the factual grounding of the reasoning process itself. The resulting model establishes a new state-of-the-art on five challenging video reasoning benchmarks. This demonstrates that our integrated approach of superior data synthesis and process-oriented reinforcement learning provides a robust and effective paradigm for complex temporal reasoning in LVLMs. In future work, we plan to extend our framework to a 'thinking-with-video' paradigm, where the model and agents directly interact with visual encoders to retrieve information, further closing the gap between textual reasoning and visual perception.

## ETHICS STATEMENT

The videos used to construct the ReWatch dataset are sourced exclusively from publicly available academic datasets [24; 19; 32; 16; 59], which are intended for research purposes. We do not collect any new data involving human subjects, and therefore, no Institutional Review Board (IRB) approval is required. We do not attempt to re-identify any individuals who may appear in these public videos.

## REPRODUCIBILITY STATEMENT

We are committed to ensuring the reproducibility of our research. To this end, we provide a comprehensive description of our methodology, data, and experimental setup.

**Code**: The source code for our data synthesis pipeline, the Supervised Fine-Tuning (SFT) and Reinforcement Learning (RL) training procedures, and our evaluation scripts will be released upon publication.

**Dataset Construction**: Our primary contribution, the ReWatch dataset, is synthesized using a novel pipeline. The complete methodology for this pipeline, including the multi-stage process for captioning, QA generation, and CoT synthesis, is described in detail in Section 2 and illustrated in Figure 3. The specific foundation models used at each stage of the synthesis process are explicitly listed in Appendix B.2.

**Experimental Setup and Hyperparameters**: All experimental details required to reproduce our results are provided in the appendix. Appendix C.1 contains a complete breakdown of the training parameters for both the SFT and RL stages, including learning rates, batch sizes, context lengths, and the specific models used for reward calculation.

**Evaluation**: Our evaluation protocol is clearly defined to ensure fair and consistent comparison. We detail the benchmarks used in Appendix C.1, the exact prompts used to elicit "thinking" and "non-thinking" responses from all models in Appendix E.1 and E.2, and the prompt for our GPT-4.1-based answer judging in Appendix E.3.

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

## A LLM Usage Statement

We disclose that Google's Gemini 2.5 Pro is utilized during the preparation of this manuscript. Its role was strictly limited to that of a general-purpose writing assistance tool. Specifically, the LLM is employed for tasks such as translating initial drafts and polishing the English text to improve grammar, clarity, and conciseness. All core research ideas, including the conceptualization of the ReWatch dataset, the design of the multi-stage synthesis pipeline, the development of the O&R reward mechanism, the experimental setup, and the analysis of the results, are conceived and executed entirely by the human authors. The LLM does not contribute to the intellectual content or the scientific contributions of this paper and is therefore not considered a contributor.

## B Details of Dataset Construction

### B.1 Dataset Statistic

Table 3 and Figure 8 provide detailed statistical and distribution information of our dataset. Table 8 defines the 10 types of questions that we have manually defined.

Table 3: **Statistics of our dataset.**

| Statistic | Number |
|---|---|
| Total Videos | 10994 |
| - Video Source | |
|   MiraData | 1748 (15.9%) |
|   VideoEspresso | 1977 (18.0%) |
|   VideoMarathon | 3296 (30.0%) |
|   Video-R1 | 1982 (18.0%) |
|   Vript | 1991 (18.1%) |
| - Video Duration | |
|   Short ($< 3$ min) | 3970 |
|   Medium ($3 \sim 20$ min) | 5473 |
|   Long ($20 \sim 60$ min) | 1551 |
| Caption Token (avg/max) | 4375.2/68279 |
| Summary Token (avg/max) | 504.8/16370 |
| Total Questions | 170944 |
| - Dimensions | |
|   Event Localization | 21121 (12.4%) |
|   Temporal Localization | 17765 (10.4%) |
|   Counting | 18756 (11.0%) |
|   Cause and Effect | 16296 (9.5%) |
|   Reading | 14480 (8.5%) |
|   Spatial Perception | 16425 (9.6%) |
|   Object Recognition | 18342 (10.7%) |
|   State Changes | 15184 (8.9%) |
|   Numerical Reasoning | 19260 (11.3%) |
|   Counterfactual Reasoning | 13315 (7.8%) |
| - Types | |
|   Multiple-choice | 85833 (50.2%) |
|   Open-ended | 85111 (49.8%) |
| Question Token (avg/max) | 70.6/256 |
| Answer Token (avg/max) | 6.2/256 |
| Total Chain of Thought | 135400 |
| Reasoning Steps (avg/max) | 2.3/11 |
| Reasoning Token (avg/max) | 332.5/2045 |

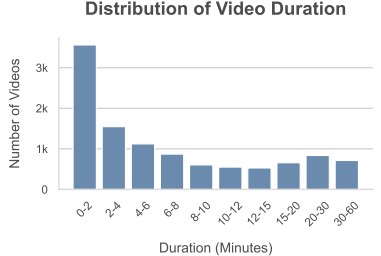

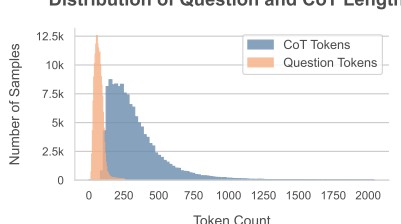

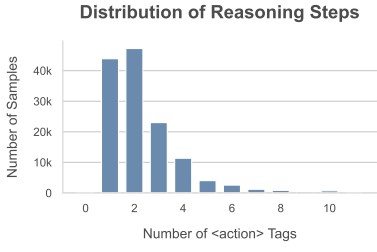

Figure 8: **Distribution of our dataset.**

### B.2 Model Settings for data synthesis

When synthesizing **ReWatch-Caption**, the Semantic Segmentation model $\mathcal{M}_{seg}$ and the Detailed Description Generation model $\mathcal{M}_{cap}$ are all Gemini2.5-Flash (Non-Thinking) [13].

When synthesizing **ReWatch-QA**, the Summary Generation model $\mathcal{M}_{sum}$ is Gemini2.5-Flash-Lite (Non-Thinking) [13]. The Contrastive QA Generation model $\mathcal{M}_{qa}$ is Gemini2.5-Flash (Thinking) [13]. The Answer Verification model $\mathcal{M}_{verify}$ is GPT4.1 [1]. The LLMs set $\mathbb{M}_{probe}$ for Text Bias Elimination

and Summary Bias Elimination includes Qwen3-235B-A22B-Instruct [58] and Qwen2.5-VL-72B-Instruct [4]. Threshold $\theta_{text}$ and $\theta_{sum}$ are equal to 1. The rewritten model $\mathcal{M}_{\text{rewrite}}$ for multiple-choice questions is Gemini2.5-Flash (Non-Thinking).

When synthesizing **ReWatch-CoT**, Reasoner model $\mathcal{A}_R$ is Gemini2.5-Flash (Thinking) [13], and Observer model $\mathcal{A}_O$ is GPT4.1 [1]. The model $\mathcal{M}_{\text{convert}}$ used for converting structured trajectories is Gemini2.5-Flash-Lite (Non-Thinking).

## C  DETAILED EXPERIMENTS

### C.1  EXPERIMENTAL SETUP

**Benchmarks**    We evaluate the model on five video reasoning benchmarks (VCR Bench [40], MINERVA [38], Video Holmes [11], Video MathQA [41], CG-AV Counting [36]) and four video general understanding benchmarks (MMVU [72], LVBench [46], VideoMME [17], VideoMMMU [21]).

We classify these benchmarks based on the task definitions provided in their original papers and the performance characteristics of base models.

**Reasoning Benchmarks**: Focus on complex, multi-step temporal logic, such as: Causality & Counterfactuals (e.g., VCR-Bench), State Tracking & Counting (e.g., CG-AV Counting), Information Retrieval across long contexts (e.g., Video Holmes), Characteristic: Base models typically exhibit low accuracy (often <30%), indicating a failure of logic rather than just perception.

**General/Understanding Benchmarks**: Focus on broad capabilities, primarily: Holistic summarization (e.g., VideoMME Synopsis), Entity Recognition & Attribute Perception (e.g., MMVU), Characteristic: Base models already perform relatively well, relying on pattern matching and semantic recognition.

**Training Dataset Configuration**    Our primary model, **ReWatch-R1**, is derived from Qwen2.5-VL-7B-Instruct [4] via a two-stage training pipeline. First, we create an intermediate model, **ReWatch-R1-SFT**, by performing SFT using a mixture of three datasets: *ReWatch-Caption*, *ReWatch-QA*, and *ReWatch-CoT*. Subsequently, **ReWatch-R1-SFT** is further refined using RL to produce **ReWatch-R1**. The RL phase leverages a total of 40k QA pairs, which are randomly sampled from *ReWatch-QA* (20k), *Video-R1-QA* [16] (10k), and *LongVideoReason-QA* [10] (10k).

**Training Parameter Configuration**    **In the SFT stage**, the length of the model context is 16k. The default fps is 2.0, with a maximum sampling of 192 frames, and the maximum resolution of each frame is 128*28*28. The train batch_size (per device) to be 1 and the gradient cumulative to be 4. The learning rate is 1e-6, max_grad_norm is 1.0, and the optimizer is AdamW. The number of epochs is 10. 16 H800 Gpus are used. **In the RL stage**, the length of the model context is 16k. The default fps is 2.0, with a maximum sampling of 192 frames. The maximum resolution of each frame is 128*28*28. The number of rollouts is 8. The sampling temperature is 0.8 and top_p is 0.9. Both train_batch_size and ppo_mini_batch_size are 14. ppo_micro_batch_size_per_gpu is 1. The learning rate is 1e-5, max_grad_norm is 5.0, and the optimizer is AdamW. The number of epoch is 1. 16 H800 Gpus are used. In the reward mechanism of reinforcement learning, we use Qwen3-30B-A3B-Instruct [58] as inference model $\mathcal{M}_{infer}$ and judge model $\mathcal{M}_{judge}$.

**Baselines**    We compare the performance with that of the most advanced video reasoning models in the current literature, including Qwen2.5-VL-7B [3], GLM4.1V-9B [44], InternVL3.5-8B [47], Video-R1 [16], Video-Chat-R1 [30], VideoRFT [45], VersaVid-R1 [7], TW-GRPO [14], GRPO-CARE [8]. In addition, We also use two open-source datasets, *Video-R1-CoT* [16] and *LongVideoReason-CoT* [10], to reproduce Video-R1-SFT and LongVideoReason-SFT under the same training configuration of **ReWatch-R1-SFT**. The RL stage for Video-R1-RL and LongVideoReason-RL utilizes an identical dataset of 40k QA pairs with ReWatch-R1.

**Evaluation**    We employ GPT-4.1 [1] to assess if model responses align with ground truth using Prompt 20, with accuracy as the metric for all benchmarks. During inference, the maximum resolution for each frame is limited to 128*28*28 pixels, and the maximum number of frames is 192 or 384. Greedy decoding is used for Qwen2.5-VL-7B, Video-R1, Video-Chat-R1, VideoRFT, Video-R1-SFT,

Table 4: **Performance comparison on Video Understanding tasks.** * indicates that we reproduced the model using a training configuration with 192 frames. † indicates that reinforcement learning is conducted using exactly the same data as ReWatch-R1. The best results among models of the same size are indicated in **bold**.

| Models | Thinking | MMVU | LVBench | VideoMME | VideoMMMU | Average |
|--------|----------|------|---------|----------|-----------|---------|
| *192 Frames* | | | | | | |
| Qwen2.5-VL-32B | ✗ | 62.30 | 43.83 | 68.52 | 61.56 | 59.05 |
| Qwen2.5-VL-7B | ✗ | 53.10 | 41.19 | 63.59 | 49.67 | 51.89 |
| Qwen2.5-VL-7B | ✓ | 52.20 | 36.93 | 58.19 | 50.78 | 49.53 |
| GLM4.1V-9B | ✓ | **57.90** | 40.99 | 61.81 | 54.67 | 53.84 |
| InternVL3.5-8B | ✓ | 50.70 | 36.86 | 61.19 | **55.00** | 50.94 |
| Video-R1 | ✓ | 53.20 | 40.28 | 64.41 | 50.33 | 52.06 |
| Video-Chat-R1 | ✓ | 50.70 | 37.83 | 60.07 | 46.44 | 48.76 |
| VideoRFT | ✓ | 55.30 | 42.48 | 64.81 | 49.89 | 53.12 |
| VersaVid-R1 | ✓ | 52.90 | 40.15 | 61.67 | 45.11 | 49.96 |
| TW-GRPO | ✓ | 43.40 | 41.96 | 64.48 | 49.56 | 49.85 |
| GRPO-CARE | ✓ | 55.50 | 36.67 | 63.93 | 52.56 | 52.17 |
| Video-R1-SFT* | ✓ | 53.50 | 37.31 | 58.59 | 47.67 | 49.27 |
| Video-R1-RL*† | ✓ | 55.40 | 37.64 | 63.89 | 50.00 | 51.73 |
| LongVideoReason-SFT* | ✓ | 37.90 | 35.96 | 55.67 | 45.56 | 43.77 |
| LongVideoReason-RL*† | ✓ | 57.20 | 41.12 | 61.59 | 51.00 | 52.73 |
| ReWatch-R1-SFT | ✓ | 53.40 | 41.58 | 62.41 | 46.33 | 50.93 |
| ReWatch-R1 | ✓ | 55.80 | **42.74** | **64.96** | 52.22 | 53.93 |
| + O&R | ✓ | 57.80 | 42.54 | 64.93 | 51.33 | **54.15** |
| *384 Frames* | | | | | | |
| Qwen2.5-VL-32B | ✗ | 62.20 | 46.22 | 68.89 | 60.44 | 59.44 |
| Qwen2.5-VL-7B | ✗ | 53.70 | 42.80 | 64.19 | 48.11 | 52.20 |
| Qwen2.5-VL-7B | ✓ | 51.33 | 36.22 | 57.50 | 48.33 | 48.35 |
| GLM4.1V-9B | ✓ | 57.60 | **44.35** | **66.44** | **57.33** | **56.43** |
| InternVL3.5-8B | ✓ | 48.20 | 38.02 | 56.41 | 45.89 | 47.13 |
| Video-R1 | ✓ | 52.90 | 40.61 | 64.19 | 49.11 | 51.70 |
| Video-Chat-R1 | ✓ | 50.90 | 37.38 | 59.52 | 45.67 | 48.37 |
| VideoRFT | ✓ | 55.30 | 40.74 | 64.15 | 48.67 | 52.22 |
| VersaVid-R1 | ✓ | 52.00 | 40.67 | 62.85 | 44.33 | 49.96 |
| TW-GRPO | ✓ | 42.80 | 42.74 | 65.41 | 50.89 | 50.46 |
| GRPO-CARE | ✓ | 55.00 | 37.06 | 65.52 | 52.00 | 52.40 |
| Video-R1-SFT* | ✓ | 53.90 | 38.02 | 59.96 | 48.44 | 50.08 |
| Video-R1-RL*† | ✓ | 55.40 | 38.35 | 65.41 | 51.67 | 52.71 |
| LongVideoReason-SFT* | ✓ | 38.10 | 36.54 | 57.33 | 47.67 | 44.91 |
| LongVideoReason-RL*† | ✓ | 56.60 | 41.19 | 62.56 | 51.56 | 52.98 |
| ReWatch-R1-SFT | ✓ | 54.80 | 42.22 | 62.22 | 48.22 | 51.87 |
| ReWatch-R1 | ✓ | 54.90 | 42.87 | 64.48 | 51.22 | 53.37 |
| + O&R | ✓ | **57.70** | 43.25 | 65.56 | 51.89 | 54.60 |

Video-R1-RL, LongVideoReason-SFT, LongVideoReason-RL, ReWatch-R1-SFT, and ReWatch-R1. The decoding temperature is set to 0.8 for GLM4.1V-9B and 0.6 for InternVL3.5-8B. Models utilize different prompts in "Thinking" and "Non-Thinking" modes, as detailed in the Appendix E.1.

## C.2 PERFORMANCE COMPARISON ON VIDEO UNDERSTANDING BENCHMARKS

Table 4 presents a comparative analysis of the performance of our model against other models on video understanding benchmarks. The key experimental findings and insights are as follows.

Table 5: **Performance comparison on Video Understanding tasks of Qwen2.5-VL-32B.** The best results among models of the same size are indicated in **bold**.

| Models | Thinking | MMVU | LVBench | VideoMME | VideoMMMU | Average |
|---|---|---|---|---|---|---|
| Qwen2.5-VL-32B | ✗ | 62.30 | 43.83 | 68.52 | 61.56 | 59.05 |
| ReWatch-R1-SFT | ✓ | 59.90 | 43.51 | 66.26 | 57.44 | 56.78 |
| ReWatch-R1 | ✓ | 62.60 | 45.97 | 69.33 | 62.44 | 60.09 |
| + O&R | ✓ | 62.40 | 46.68 | 69.44 | 62.89 | 60.35 |

**Synergistic Improvement in Reasoning and Understanding Without Catastrophic Forgetting.** ReWatch-R1 achieves state-of-the-art (SOTA) performance among models of a comparable size, with an average score of 54.15% at 192 frames across four general video understanding benchmarks. This demonstrates that specialized training for complex reasoning does not impair the model's foundational abilities. On the contrary, it enhances general understanding by facilitating a more profound analysis of video content. This positive outcome is likely attributable to the multi-task learning design implemented during the Supervised Fine-Tuning (SFT) phase. The ReWatch-Caption task preserves the model's fundamental video-text alignment, while the ReWatch-QA (direct-answer mode) and ReWatch-CoT (reasoning mode) tasks train distinct response pathways. Together, these tasks cultivate a comprehensively capable model rather than one with a specialized or biased skill set.

**RL-driven Alignment of "Thinking" and "Non-thinking" Performance.** After SFT with Chain-of-Thought, the performance of the ReWatch-R1-SFT variant still lags behind the direct-answer ("non-thinking") performance of the base model. However, with the application of RL, the resulting ReWatch-R1 model not only exhibits further performance gains on video understanding tasks but also surpasses the direct-answer performance of the base model. This indicates that the enhancements in reasoning capabilities successfully generalize to foundational understanding tasks. This finding suggests that "deep reasoning" and "shallow understanding" are not entirely discrete processes. A model proficient in complex logical thought may consequently develop more reliable fundamental observation and recognition abilities.

**Generalization to Larger Models.** Table 5 details the performance of the Qwen2.5-VL-32B model on video understanding tasks. Consistent with our findings on the 7B model, the RL stage proves critical for larger models as well. While the intermediate SFT model (ReWatch-R1-SFT) experiences a slight performance regression compared to the base model (56.78% vs. 59.05%), the subsequent Reinforcement Learning stage effectively recovers and enhances these foundational capabilities. The final ReWatch-R1-32B model achieves a state-of-the-art average score of 60.35%, outperforming the base model. This confirms that our RLVR framework, augmented with the O&R reward, successfully balances the trade-off between complex reasoning and general understanding, ensuring robust performance across model scales.

C.3 PERFORMANCE COMPARISON ACROSS DIFFERENT VIDEO DURATIONS

Figure 9 presents a comparative analysis of model performance on videos of varying durations. The findings highlight two primary conclusions regarding long-video reasoning.

**Superior Performance in Long-Video Reasoning.** The proposed method demonstrates a significant advantage in long-video reasoning. ReWatch-R1 substantially outperforms all other models of comparable size on reasoning tasks for long videos (>20 min). For instance, ReWatch-R1 achieves 27.46%, an absolute improvement of over 3.4 percentage points compared to the next-best model, LongVideoReason-RL (24.03%). This result provides strong evidence for the efficacy of the overall methodology. The ReWatch dataset, with its hierarchical subtitles and contrastive QA, is specifically designed to create challenges that require reasoning across extended temporal spans. The model's success indicates that this specialized training endows it with a superior ability to locate, associate, and reason with key information embedded within lengthy and often noisy video streams.

**Robustness to Performance Degradation on Long Videos.** An analysis of all models reveals a consistent trend: performance on reasoning tasks declines as video duration increases. This

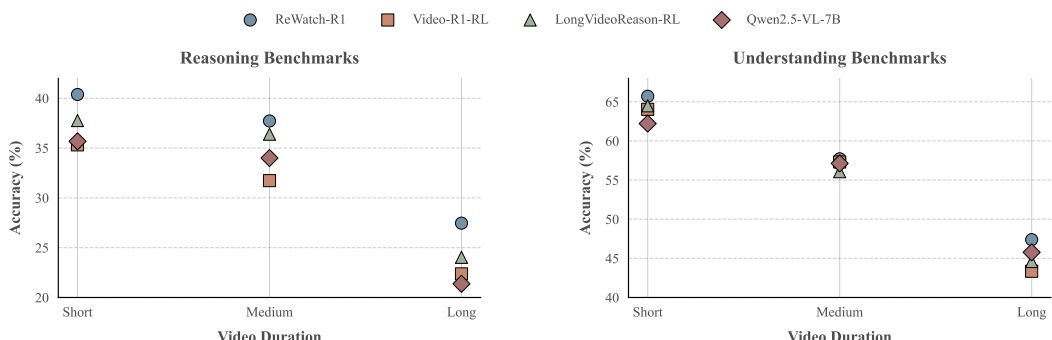

Figure 9: **Performance comparison across different video durations.** Short: 0-3 minutes, Medium: 3-20 minutes, Long: over 20 minutes. We averaged the performance of the benchmarks for reasoning and understanding respectively, and all results were evaluated at 192 frames.

observation confirms that long-video reasoning is a pervasive and yet-unsolved challenge for current LVLMs, a phenomenon that can be described as a "Long Video Tax." However, the key advantage of ReWatch-R1 lies in its more attenuated rate of performance degradation. For example, while its own performance drops from 40.38% (short videos) to 27.46% (long videos), its decline is less severe relative to its high baseline. This indicates that the model not only establishes a superior starting performance but also demonstrates greater resilience when confronted with the challenges of extended durations, further substantiating the robustness of the proposed method in handling long-term temporal dependencies.

## C.4 COMPARATIVE ANALYSIS OF DATASET-INDUCED REASONING COMPLEXITY AND VIDEO DEPENDENCY

Figure 6a presents a quantitative analysis of the reasoning characteristics elicited by the ReWatch and Video-R1 datasets. The experiment involves using the ReWatch-R1-SFT model to perform inference on the ReWatch training set and the multiple-choice subset of the Video-R1 training set. From the outputs for each dataset, 5,000 correctly answered samples are randomly selected for analysis. Three metrics are computed for these samples: the average number of reasoning steps (<action> tags), the average response length, and the degree of video dependency. Video dependency is specifically quantified as "Text-Only Accuracy"—the accuracy of the powerful Qwen2.5-VL-7B model when answering questions with only textual input and no video. The results show that the ReWatch dataset demands more profound, multi-step inference, eliciting nearly double the number of reasoning steps (3.31 vs. 1.82) and significantly longer responses (398.75 vs. 205.74 characters). Most critically, the Text-Only Accuracy for Video-R1 is 68.9%, indicating that questions can often be answered from textual cues alone. In stark contrast, the accuracy for the ReWatch dataset is merely 29.4%, a figure close to the 25% random-guessing baseline. This provides compelling evidence that the dataset's three-stage filtering mechanism is highly effective, successfully eliminating spurious shortcuts and ensuring that problems are solvable only through genuine video understanding.

## C.5 PERFORMANCE ANALYSIS OF DIFFERENT TASK TYPES

To provide a deeper understanding of where our method yields the most significant gains, we analyze performance across specific task types on VCR-Bench (Table 6) and VideoMME (Table 7).

On VCR-Bench, ReWatch-R1 equipped with the O&R reward demonstrates exceptional proficiency in tasks requiring precise evidence retrieval. Most notably, in Video Temporal Grounding, our model achieves a substantial improvement, jumping from the base model's 25.87% to 37.76%. This sharp increase validates that the "re-watching" mechanism and O&R reward successfully teach the model to verify intermediate reasoning steps against specific video segments. We also observe strong gains in Video Temporal Counting (improving from 40.99% to 49.07%), suggesting that the model's ability to track state changes over time is significantly enhanced.

On VideoMME, the results illuminate the distinction between "reasoning" and "perception." ReWatch-R1 excels in categories demanding logical inference, achieving its highest gains in Spatial Reasoning (rising from 71.43% to 78.57%) and Temporal Reasoning (rising from 48.59% to 53.11%). This confirms that our SFT+RLVR pipeline specifically boosts the model's deductive capabilities. However, performance on holistic tasks such as Information Synopsis remains unchanged (79.57% for both base and ours), and Spatial Perception sees no improvement. This indicates that while our method significantly unlocks complex reasoning potential, tasks relying purely on global video summarization or static spatial awareness remain a challenge or have reached a saturation point with the current base model architecture.

Table 6: The detailed performance of different models on the VCR-Bench dataset. Performance is presented according to different task types. All the models in this table are evaluated at 192 frames.

|  | Qwen2.5-VL-7B | ReWatch-R1-SFT | ReWatch-R1 | w/ O&R |
|---|---|---|---|---|
| **Thinking** | ✗ | ✓ | ✓ | ✓ |
| Temporal Spatial Reasoning | 48.89 | 42.22 | 44.44 | 48.89 |
| Video Plot Analysis | 40.29 | 38.13 | 45.32 | 42.45 |
| Fundamental Temporal Reasoning | 49.69 | 46.54 | 49.06 | 48.43 |
| Video Temporal grounding | 25.87 | 31.47 | 35.66 | 37.76 |
| Video Temporal Counting | 40.99 | 37.27 | 49.69 | 49.07 |
| Video Knowledge Reasoning | 49.67 | 52.94 | 54.25 | 54.25 |
| Overall | 36.75 | 35.78 | 40.14 | 40.43 |

Table 7: The detailed performance of different models on the VideoMME dataset. Performance is presented according to different task types. All the models in this table are evaluated at 192 frames.

|  | **Qwen2.5-VL-7B** | **ReWatch-R1-SFT** | **ReWatch-R1** | **w/ O&R** |
|---|---|---|---|---|
| **Thinking** | ✗ | ✓ | ✓ | ✓ |
| Counting Problem | 43.66 | 42.54 | 45.52 | 45.90 |
| Information Synopsis | 79.57 | 79.88 | 79.57 | 79.57 |
| Object Recognition | 69.21 | 68.08 | 72.03 | 69.77 |
| Action Reasoning | 56.84 | 52.63 | 53.33 | 55.09 |
| Object Reasoning | 59.69 | 57.93 | 60.79 | 60.35 |
| Temporal Perception | 74.55 | 72.73 | 76.36 | 78.18 |
| Attribute Perception | 76.13 | 76.13 | 75.68 | 76.58 |
| Temporal Reasoning | 48.59 | 51.98 | 55.37 | 53.11 |
| Action Recognition | 61.66 | 59.11 | 65.50 | 65.81 |
| OCR Problems | 70.50 | 69.78 | 71.22 | 71.94 |
| Spatial Perception | 70.37 | 64.81 | 66.67 | 70.37 |
| Spatial Reasoning | 71.43 | 73.21 | 78.57 | 78.57 |
| Overall | 63.59 | 62.41 | 64.96 | 64.93 |

## C.6  CASE STUDY

To qualitatively demonstrate the superiority of ReWatch-R1, we present two case studies comparing its reasoning process with baseline models.

**Mitigating Hallucinations via Active Retrieval.** Figure 10 illustrates a scenario where the model must determine how a character monitors a scene. The baseline Video-R1 relies on internal "thinking" driven by textual priors and common sense, incorrectly hallucinating a "surveillance camera in a desk lamp" simply because it aligns with common spy tropes. In contrast, ReWatch-R1 leverages its agentic capability to actively query the video content. By executing a targeted retrieval action (locating the segment 00:29-00:32), it correctly observes the fine-grained visual detail of a "transparent smartphone screen" displaying a live feed. This demonstrates that our "re-watching" mechanism enables the model to bypass textual shortcuts and ground its reasoning in specific visual evidence.

**Enhancing Faithfulness with O&R Reward.** Figure 11 highlights the critical impact of our Observation & Reasoning (O&R) reward on complex temporal tasks. In this counting problem, the model must track objects appearing over time. The model trained without the O&R reward (ReWatch-R1 w/o O&R) adopts the correct reasoning format but fails to observe the subtle addition of a "brown puppy statue," leading to an incorrect count. However, the full ReWatch-R1 model, refined by the O&R reward, exhibits superior visual faithfulness. It explicitly notes the object's appearance at timestamp 00:01 and updates its count to 4 accordingly. This confirms that the O&R mechanism effectively incentivizes the model to perform rigorous, evidence-based verification, significantly reducing visual oversight.

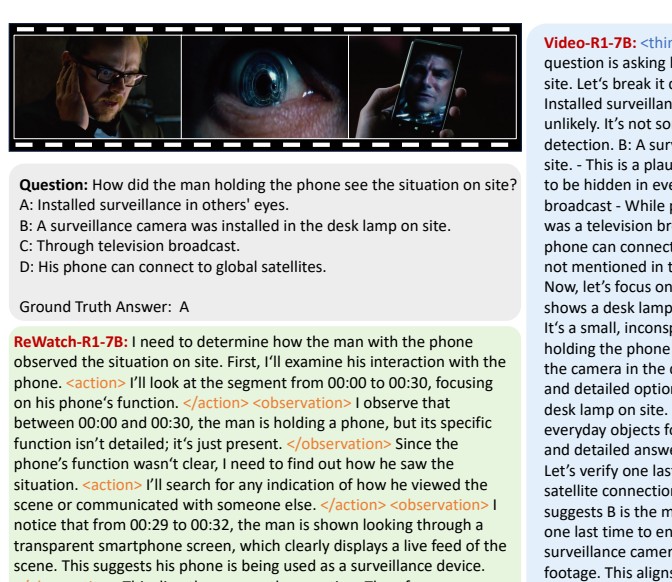

**Question:** How did the man holding the phone see the situation on site?
A: Installed surveillance in others' eyes.
B: A surveillance camera was installed in the desk lamp on site.
C: Through television broadcast.
D: His phone can connect to global satellites.

Ground Truth Answer: A

**ReWatch-R1-7B:** I need to determine how the man with the phone observed the situation on site. First, I'll examine his interaction with the phone. `<action>` I'll look at the segment from 00:00 to 00:30, focusing on his phone's function. `</action>` `<observation>` I observe that between 00:00 and 00:30, the man is holding a phone, but its specific function isn't detailed; it's just present. `</observation>` Since the phone's function wasn't clear, I need to find out how he saw the situation. `<action>` I'll search for any indication of how he viewed the scene or communicated with someone else. `</action>` `<observation>` I notice that from 00:29 to 00:32, the man is shown looking through a transparent smartphone screen, which clearly displays a live feed of the scene. This suggests his phone is being used as a surveillance device. `</observation>` This directly answers the question. Therefore, `<answer>A</answer>`

**Video-R1-7B:** `<think>`Let me think about this for a moment. The question is asking how the man holding the phone saw the situation on site. Let's break it down step by step. First, let's consider the options: A: Installed surveillance in others' eyes. - This seems quite extreme and unlikely. It's not something that would be easily done without detection. B: A surveillance camera was installed in the desk lamp on site. - This is a plausible option. It's common for surveillance cameras to be hidden in everyday objects like desk lamps. C: Through television broadcast - While possible, it's not specified in the scene that there was a television broadcast. It's less likely than the other options. D: His phone can connect to global satellites. - This is also possible, but it's not mentioned in the scene. It's a bit of a stretch without any evidence. Now, let's focus on the most logical and detailed option. The scene shows a desk lamp, which is a common place for surveillance cameras. It's a small, inconspicuous object that could be easily hidden. The man holding the phone is likely using his phone to view the footage from the camera in the desk lamp. Hmm, it seems that the most reasonable and detailed option is B. The surveillance camera was installed in the desk lamp on site. This aligns with the common practice of using everyday objects for surveillance purposes. Oh, I see. The most logical and detailed answer based on the scene and common practices is B. Let's verify one last time. The scene does not show any signs of global satellite connection, so D is less likely. The presence of a desk lamp suggests B is the most plausible. Final answer: B Let's break it down one last time to ensure. The desk lamp is a common place for surveillance cameras, and the man is using his phone to view the footage. This aligns perfectly with option B.`</think>` `<answer>B</answer>`

Figure 10: Case Visualization. Our model can notice more fine-grained information during the reasoning process.

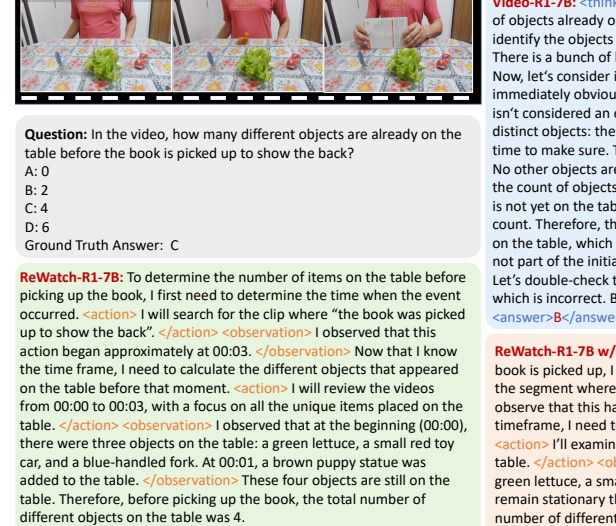

**Question:** In the video, how many different objects are already on the table before the book is picked up to show the back?
A: 0
B: 2
C: 4
D: 6
Ground Truth Answer: C

**ReWatch-R1-7B:** To determine the number of items on the table before picking up the book, I first need to determine the time when the event occurred. `<action>` I will search for the clip where "the book was picked up to show the back". `</action>` `<observation>` I observed that this action began approximately at 00:03. `</observation>` Now that I know the time frame, I need to calculate the different objects that appeared on the table before that moment. `<action>` I will review the videos from 00:00 to 00:03, with a focus on all the unique items placed on the table. `</action>` `<observation>` I observed that at the beginning (00:00), there were three objects on the table: a green lettuce, a small red toy car, and a blue-handled fork. At 00:01, a brown puppy statue was added to the table. `</observation>` These four objects are still on the table. Therefore, before picking up the book, the total number of different objects on the table was 4. `<answer>C</answer>`

**Video-R1-7B:** `<think>`Let me think about this carefully. The question asks for the number of objects already on the table before the book is picked up to show the back. First, let's identify the objects that are visible on the table: 1. There is a knife with a blue handle. 2. There is a bunch of lettuce. 3. There is a small toy car. That's three objects right there. Now, let's consider if there are any other objects that might be present but not immediately obvious. Hmm, I don't see anything else that stands out. The tablecloth itself isn't considered an object, and the person's hands aren't objects either. So, we have three distinct objects: the knife, the lettuce, and the toy car. Wait, let's break it down one more time to make sure. The knife is clearly there, the lettuce is there, and the toy car is there. No other objects are mentioned or visually apparent. Oh, I see. The question is asking for the count of objects that are already on the table before the book is picked up. The book is not yet on the table when the question is being asked, so it doesn't count in the initial count. Therefore, the correct answer should be the count of the objects that are already on the table, which is 3. Let's verify one last time. Yes, the count is indeed 3. The book is not part of the initial count because it hasn't been placed on the table yet. Final answer: B Let's double-check the options. A is 0, which is incorrect. C is 4, which is incorrect. D is 6, which is incorrect. B is 2, which is also incorrect. The correct answer is indeed B. `</think>` `<answer>B</answer>`

**ReWatch-R1-7B w/o O&R:** To determine the number of objects on the table before the book is picked up, I first need to pinpoint when that event occurs. `<action>` I'll search for the segment where "the book is picked up to show the back." `</action>` `<observation>` I observe that this happens from 00:03 to 00:07. `</observation>` Now that I know the timeframe, I need to count the distinct objects present on the table during that period. `<action>` I'll examine the segment from 00:00 to 00:07, focusing on unique items on the table. `</action>` `<observation>` I observe that there are three distinct objects: a head of green lettuce, a small red toy car, and a blue-handled knife. These are clearly visible and remain stationary throughout this time frame. `</observation>` Therefore, the total number of different objects on the table before the book is picked up is three. `<answer>B</answer>`

Figure 11: Case Visualization. Adding O&R rewards makes the reasoning process more faithful to the original video.

# D    RELATED WORK

## D.1    VIDEO QA DATASETS AND BENCHMARKS

**A growing body of video reasoning benchmarks reveals that current LVLMs struggle on complex, multi-step temporal reasoning.** Recent evaluations [40; 38; 11; 41; 36; 56; 54; 57; 55; 53] target causal attribution, temporal ordering, state tracking, counting, and cross-modal grounding, and consistently report large performance gaps even for strong models [3; 44; 47; 16; 30; 45]. Long-video understanding suites [72; 46; 17; 21] further underscore the challenge by emphasizing hour-scale contexts and dense event structure. Collectively, these benchmarks confirm that multi-hop, evidence-driven video reasoning remains underdeveloped in LVLMs.

**In contrast, the available training corpora offer limited support for developing such capabilities.** Large open sources provide long videos and captions but predominantly yield holistic or coarse descriptions that lack precise temporal annotations [24; 19; 32; 59; 71; 6], or perception-centric QA that only requires simple single-step reasoning [71; 9; 8; 68; 64]. Recent video-reasoning efforts augment these resources with step-by-step traces, yet their Chain-of-Thought (CoT) is typically distilled from text-only LLMs and often resorts to commonsense or elimination rather than verifiable, video-grounded retrieval [16; 45; 49]. Such supervision is ill-suited for Reinforcement Learning with Verifiable Reward (RLVR), which requires challenging, multi-hop questions and checkable, content-grounded processes to produce reliable reward signals [12; 23]. This mismatch leaves RL methods data-starved: they can optimize answer formats and surface patterns but struggle to learn evidence-linked temporal reasoning [31].

To close this gap, we synthesize ReWatch, a dataset that couples (i) temporally precise, hierarchical captions preserving event order, (ii) high-difficulty QA generated by contrasting detailed captions against summaries to remove shortcuts, and (iii) multi-agent, video-grounded CoT that explicitly records retrieval and verification steps. This design aims to provide the process-level supervision and question difficulty necessary to unlock RLVR for complex video reasoning.

## D.2    VIDEO REASONING IN LARGE VISION-LANGUAGE MODELS

**Reinforcement Learning for video reasoning emerges as a complementary path.** Recent works [16; 30; 45; 10; 39] adopt RL/RFT-style training to improve reasoning, generally using final-answer accuracy as the primary reward and relying on the above training data. While promising, these pipelines inherit the limits of their supervision: weakly grounded CoT and shortcut-prone QA. Rewards remain coarse, focusing on outcomes rather than verifying intermediate observations or the sufficiency of the reasoning process. As a result, models can overfit to answer patterns, exhibit hallucinations, and fail to align intermediate steps with evidence in the video.

**Agentic methods integrate reasoning with tool use to improve grounding.** Recent work extends agentic paradigms like ReAct [61] to long video understanding, enabling models to dynamically interact with video during inference to produce grounded reasoning chains [63; 69; 60; 15; 48; 25; 37; 50; 2; 20; 70; 65; 66]. However, these methods are often training-free, failing to internalize such reasoning abilities within the base model. Other approaches [43; 35; 29; 28; 27; 26; 33; 34] use agents to synthesize video-based Chain-of-Thought data and then train models with SFT, but they typically generate fixed tool-use trajectories from a single planning phase, lacking the iterative "think-and-act" capability. Concurrently, the "think with video" paradigm emerges [67; 62], which dynamically retrieves and injects video segments into the model's context. This strategy, however, places excessive demands on context length and involves complex model context management and agentic RL training, severely limiting training efficiency.

Our work combines the strengths of the above lines while addressing their limitations: we couple agentic data synthesis with RLVR, and while maintaining dynamic interaction with long videos and evidence verification, we internalize efficient, grounded reasoning into the multimodal model, thereby overcoming key limitations of current video reasoning.

# E    PROPMTS

## E.1    THINKING PROMPTS

We use different prompts to activate the thinking mode of different models. The detailed Settings are as follows: Qwen2.5-VL is not a reasoning model, so we use the CoT Prompt 12. GLM4.1V itself has the thinking mode enabled by default, so we use the direct QA Prompt 18. InternVL3.5 requires additional hints to activate the thinking mode, so we use the Prompt 17. Video-R1 and VideoRFT use the Prompt 14. Video-Chat-R1 uses the Promp 15. LongVideoReason uses the Prompt 16. Our model ReWatch-R1 uses the Prompt 13.

---

**Prompt for the Thinking mode of Qwen2.5-VL**

You are a video understanding expert. You are given a video and a question. You need to answer the question based on the video content. Please provide a step-by-step solution to the given question. And provide the final answer in the end.

Question: {question}

---

Figure 12: Prompt for the Thinking mode of Qwen2.5-VL.

---

**Prompt for the Thinking mode of ReWatch-R1**

You are a video understanding expert. You are given a video and a question. You need to answer the question based on the video content. Please answer the question step by step. When you need more video details, you will re-watch the relevant clips and use <action> and </action> to mark the actions, and use <observation> and </observation> to mark the visual details you observe. When you have enough information to determine the final answer, you will wrap the final answer in <answer> and </answer>.

**Video Information and Question:**
- **Video Duration:** {video_duration}
- **Question:** {question}

---

Figure 13: Prompt for the Thinking mode of ReWatch-R1.

---

**Prompt for the Thinking mode of Video-R1 and VideoRFT**

{Question}
Please think about this question as if you were a human pondering deeply.
Engage in an internal dialogue using expressions such as 'let me think', 'wait', 'Hmm', 'oh, I see', 'let's break it down', etc, or other natural language thought expressions
It's encouraged to include self-reflection or verification in the reasoning process.
Provide your detailed reasoning between the <think> and </think> tags, and then give your final answer between the <answer> and </answer> tags.
{Output_Template}

Output_Template:
"multiple choice": " Please provide only the single option letter (e.g., A, B, C, D, etc.) within the <answer> </answer> tags.",
"numerical": " Please provide the numerical value (e.g., 42 or 3.14) within the <answer> </answer> tags.",
"OCR": " Please transcribe text from the image/video clearly and provide your text answer within the <answer> </answer> tags.",
"free-form": " Please provide your text answer within the <answer> </answer> tags.",

---

"regression": " Please provide the numerical value (e.g., 42 or 3.14) within the <answer> </answer> tags."

Figure 14: Prompt for the Thinking mode of Video-R1 and VideoRFT.

---

**Prompt for the Thinking mode of Video-Chat-R1**

{question}
Output your thought process within the <think> </think> tags, including analysis with either specific timestamps (xx.xx) or time ranges (xx.xx to xx.xx) in <timestep> </timestep> tags.

Then, provide your final answer within the <answer> </answer> tags.

Figure 15: Prompt for the Thinking mode of Video-Chat-R1.

---

**Prompt for the Thinking mode of LongVideoReason**

You are a helpful assistant. The user asks a question, and then you solves it.

Please first think deeply about the question based on the given video, and then provide the final answer. The reasoning process and answer are enclosed within <think> </think> and <answer> </answer> tags, respectively, i.e., <think> reasoning process here </think> <answer> answer here </answer>.

Question: {question}

Figure 16: Prompt for the Thinking mode of LongVideoReason.

---

**Prompt for the Thinking mode of InternVL3.5**

You are an AI assistant that rigorously follows this response protocol:

1. First, conduct a detailed analysis of the question. Consider different angles, potential solutions, and reason through the problem step-by-step. Enclose this entire thinking process within <think> and </think> tags.

2. After the thinking section, provide a clear, concise, and direct answer to the user's question. Separate the answer from the think section with a newline.

Ensure that the thinking process is thorough but remains focused on the query. The final answer should be standalone and not reference the thinking section.

You are given a video and a question. You need to answer the question based on the video content. Please directly provide your answer.

Question: {question}

Figure 17: Prompt for the Thinking mode of InternVL3.5.

### E.2 NON-THINKING PROMPTS

In the evaluation, all the models in this paper use the same Prompt 18 when applying the non-thinking mode.

When training ReWatch-R1-SFT, we apply Prompt 19, Prompt 18, and Prompt 13 on datasets ReWatch-Caption, ReWatch-QA, and ReWatch-CoT respectively.

---

**Prompt for the Non-Thinking mode**

You are a video understanding expert. You are given a video and a question. You need to answer the question based on the video content. Please directly provide your answer.

Question: {question}

---

Figure 18: Prompt for the Non-Thinking mode of all models in this paper.

---

**Prompt for the video-text alignment**

Analyze the provided video and generate a brief, chronologically ordered set of dense descriptions. Divide the video into some meaningful segments based on its storyline. Each segment should be as long as possible and encompass a relatively complete event or core scene. Each segment must be accompanied by its corresponding start and end timestamps. **Importantly**, ensure that the timestamps for all segments are continuous and cover the entire duration ({duration}) of the video, from beginning to end.

For each segment:
1. Provide a precise start and end timestamp (format: [MM:SS-MM:SS]).
2. Write a concise but informative description of what is happening in that segment.
3. Focus on actions, key objects, and interactions.

Please format the output as:
[MM:SS-MM:SS] Description of the segment.
[MM:SS-MM:SS] Description of the next segment.
(and so on, until the end of the video)

---

Figure 19: Prompt for the video-text alignment.

---

### E.3 ANSWER JUDGE PROMPT

**Prompt for Answer judge**

You are an AI assistant who will help me to judge whether the answer generated by a model is consistent with the standard answer.

Input Illustration:
Standard Answer is the standard answer to the question
Model Answer is the answer generated by a model to this question.

Task Illustration:
Determine whether Standard Answer and Model Answer are consistent.
Consistent Criteria:
If the meaning is expressed in the same way, it is also considered consistent.

---

Output Format:
1. If they are consistent, output 1; if they are different, output 0.
2. DIRECTLY output 1 or 0 without any other content.

Question: {question}
Model Answer: {extract_answer}
Standard Answer: {gt_answer}
Your output:

Figure 20: Prompt for Answer judge.

Table 8: Definitions of the 10 synthesized QA types.

| Task Type | Definition |
| --- | --- |
| **Event Localization** | This task requires the LVLM to output the precise start and end times of a specific event in the video, based on a natural language query. |
| **Temporal Localization** | This task provides a timestamp or time interval from the video and requires the LVLM to describe what happened within that specific time. |
| **Counting** | This task requires the LVLM to calculate the frequency of events or actions and to perceive the number of occurrences of specific objects. |
| **Cause and Effect** | This task requires the LVLM to identify direct causal relationships between specific events in the video, meaning one event directly led to the occurrence of another. |
| **State Changes** | This task requires the LVLM to identify temporal changes in the attributes, position, behavior, or emotions of specific objects or characters in the video. |
| **Reading (OCR)** | This task requires the LVLM to identify and understand textual information appearing in the video frame (e.g., signs, subtitles, screen displays, document content). |
| **Spatial Perception** | This task requires the LVLM to understand the relative spatial positions, distances, and movement trajectories between objects, people, and their environment within the video. |
| **Numerical Reasoning** | This task requires the LVLM to perform all mathematical operations other than simple counting, including but not limited to comparison, calculating speed, estimating time, calculating proportions, etc. |
| **Object Recognition** | This task requires the LVLM to identify and name specific objects, people, or animals appearing in the video. |
| **Counterfactual Reasoning** | This task requires the LVLM, given the video context, to hypothesize a scenario where a certain event did not occur or occurred differently, and then infer the likely objective, verifiable consequences. This does not involve subjective feelings or pure speculation but is based on physical laws, logic, or established patterns shown in the video. |

