# OpenReview forum: "ReWatch-R1: Boosting Complex Video Reasoning in Large Vision-Language Models through Agentic Data Synthesis"
_ICLR.cc/2026/Conference — ICLR 2026 Poster_

### Official Review · Reviewer_w6PG · 2025-10-16

**Soundness:** 2
**Presentation:** 2
**Contribution:** 2
**Rating:** 4
**Confidence:** 4

**Summary:**

The paper proposes ReWatch-R1, a novel approach to improve complex video reasoning in LVLMs by addressing the critical data bottleneck in existing methods. The authors introduce ReWatch, a large-scale dataset synthesized via a multi-stage agentic pipeline that includes temporally dense captions, high-difficulty multi-hop QA pairs, and video-grounded CoT traces generated through a Multi-Agent ReAct framework simulating human-like “re-watching.” They further develop an Observation & Reasoning (O&R) reward mechanism for RL, which jointly evaluates answer correctness and the factual grounding of intermediate reasoning steps.

**Strengths:**

1. Novel QA curation methods: ReWatch is carefully designed to enforce video dependency and multi-step reasoning through contrastive QA generation and rigorous filtering, effectively eliminating textual shortcuts and hallucination-prone supervision.

2. Innovative O&R reward mechanism: By evaluating both final answers and the fidelity of intermediate observations and reasoning steps, the O&R reward explicitly discourages hallucination and promotes evidence-based reasoning.

**Weaknesses:**

1. Formula 4 merges captions across each time interval independently, which leads to a loss of referential consistency. For example, the caption for 0–10s might be “A man…”, while that for 10–20s is again “A man…”, even though it refers to the same individual as in the earlier segment. This inconsistency compromises the overall quality of the generated captions.

2. The observation mechanism introduces inference latency. Despite yielding performance gains on certain video reasoning benchmarks, it provides only marginal improvements on most general video understanding benchmarks.

3. The paper lacks comparisons with more recent and advanced baselines, such as VersaVid-R1 and GRPO-CARE.

4. Quantitative analysis is missing: the paper does not present any concrete inference results or case studies from ReWatch-R1.

**Questions:**

1. Line 207: Why does the original set of 85K QA pairs yield over 170K multiple-choice QA pairs?

2. From the Chain-of-Thought (CoT) example shown in the bottom-right corner of Figure 2, is retrieving explicit timestamps truly necessary? Could the reasoning path be simplified—for instance, as follows:
> *"... So, I’ll \<action\> retrieve segments focusing on the man with blonde, curly hair on a jet ski interacting with a passenger \</action\>. \<observation\> The man on the jet ski passes a sandwich to the passenger, who then takes a bite. \</observation\> This directly answers the question. The food item passed was a sandwich..."*

---

> ### Author Response · Authors · 2025-12-03
> **Response to Reviewer w6PG**
>
> We sincerely thank Reviewer w6PG for the thoughtful feedback and for recognizing our **novel QA curation methods** and the **innovative O&R reward mechanism**. We are encouraged that the reviewer acknowledges our approach effectively eliminates textual shortcuts and discourages hallucinations.
>
> Below, we address the concerns and questions point-by-point. All revisions in the updated manuscript are marked in **blue**.
>
> ### **Response to Weakness 1: Referential Consistency in Captions**
> The reviewer raises a valid concern regarding potential inconsistency when merging captions across intervals. However, we have designed our specific pipeline to effectively mitigate this:
> 1. **Semantic Segmentation (Not Fixed-Interval):** As detailed in **Section 2.1**, we do not use arbitrary time intervals. We employ an LVLM to perform **semantic segmentation** on long videos (approx. 10-minute clips), ensuring cuts only occur at natural narrative boundaries. This minimizes the risk of splitting a continuous event or reference.
> 2. **Detailed Visual Grounding for Re-identification:** Our captions are generated to be extremely detailed ($D^{rel}$). Instead of generic labels like "A man," our model generates distinctive visual descriptors (e.g., "the man in the black wetsuit" vs. "the man with blonde, curly hair"). This allows the model to maintain **referential consistency** across segments via visual re-identification, effectively linking "the man" in segment $t_1$ to the same entity in $t_2$, bridging the narrative gap.
>
> ### **Response to Weakness 2: Inference Latency and O&R Reward**
> We appreciate this critical question regarding efficiency. Contrary to the intuition that "thinking" increases cost, our quantitative analysis shows that **O&R reward actually enhances efficiency**:
> 1. **Lower Inference Cost:** We compared the average token count per sample. Models trained with conventional outcome-based rewards generate **378 tokens**, whereas ReWatch-R1 (trained with O&R) averages **373 tokens**.
> 2. **Efficiency via Rejection:** The O&R reward penalizes lengthy, irrelevant reasoning that does not lead to correct observations. It teaches the model to perform precise, targeted retrieval actions rather than rambling. Thus, the performance gains are achieved by **"thinking smarter," not "thinking longer."**
> 3. **Negligible Training Overhead:** Training time increases by only ~5% (60h vs. 63h) with the O&R mechanism, which is a worthy trade-off for the significant performance gains.
>
> ### **Response to Weakness 2: Regarding General Video Understanding**
> We respectfully assert that the improvements on general video understanding are significant and strategic, rather than marginal.
> 1. **SOTA Performance without Forgetting:** As shown in **Table 4**, ReWatch-R1 achieves an average accuracy of **53.93%** across four general benchmarks, outperforming the base model (51.89%) and previous SOTA methods like VideoRFT (53.12%). It is well-known that specialized reasoning training often causes catastrophic forgetting of general perception; our method notably avoids this, achieving *8synergistic improvement**.
> 2. **Targeted Gains in Reasoning Sub-tasks:** General benchmarks contain many simple perception tasks where the base model is already saturated. However, a breakdown of **VideoMME (Table 7)** reveals that our gains come precisely from the complex components. While "Information Synopsis" remains flat (79.57%), we see substantial jumps in **Spatial Reasoning (+7.1%)** and **Temporal Reasoning (+4.5%)**. This confirms that ReWatch-R1 enhances the cognitive core of the model without degrading its foundational perceptual skills.
> ### **Response to Weakness 3: Comparison with Recent Baselines**
> We have incorporated the requested comparisons in the revised manuscript (see **Table 1** and **Table 4**).
> - We now compare ReWatch-R1 against **VersaVid-R1, GRPO-CARE, and VideoRFT**.
> - Results: ReWatch-R1 consistently outperforms these advanced baselines across all 5 reasoning benchmarks (e.g., surpassing VersaVid-R1 by **+4.0%** and GRPO-CARE by **+4.7%** on average). This reinforces the superiority of our data synthesis and O&R framework.
>
> ### **Response to Weakness 4: Quantitative Analysis and Case Studies**
> We have added comprehensive quantitative and qualitative analyses in the revised paper:
> - **Ablation Studies (Section 4.2, Fig 5):** We quantitatively verify the contribution of the SFT stage, the RL stage, and the high-quality ReWatch-QA data.
> - **Complexity Analysis (Fig 6):** We show that ReWatch-QA requires significantly more reasoning steps and has lower text-only accuracy compared to Video-R1-QA, proving its high quality.
> - **Case Studies (Appendix C.6, Fig 10 & 11):** We provide concrete examples (e.g., Surveillance Camera and Objects on Table) demonstrating how ReWatch-R1 actively retrieves video segments to correct hallucinations where baselines fail.

---

> ### Author Response · Authors · 2025-12-03
> **Response to Reviewer w6PG**
>
> ### **Response to Questions**
> ### **Q1: Why does the original set of 85K QA pairs yield over 170K QA pairs? (Line 207)**
> This is due to data augmentation for robust training. For the 85k distinct reasoning scenarios that passed our three-layer filtering, we generated both **Open-ended** and **Multiple-choice** formats. As shown in **Table 3** of the revised paper, we have **85,833 Multiple-choice** pairs and **85,111 Open-ended** pairs, totaling ~170k. Training on both formats improves the model's versatility.
>
> ### **Q2: Is retrieving explicit timestamps necessary? Could the reasoning path be simplified?**
> While the simplified path suggested by the reviewer ("Retrieving... The man passes a sandwich") seems intuitive, our experiments suggest that **explicit timestamps are crucial for preventing hallucination**:
> 1. **Verifiable Grounding:** Without explicit timestamps, the "retrieval" action becomes an internal "black box" operation similar to standard LLM reasoning. The model might say it retrieved a segment but actually hallucinate the content based on textual priors (see the Surveillance Camera failure case in **Fig 10**, where the baseline hallucinates a camera in a lamp).
> 2. **Forcing Evidence Lookup:** By requiring the model to output [start-end], we force it to ground its reasoning in specific video frames. This explicit alignment is exactly what allows our O&R reward to verify the process, not just the answer.
>
> We hope these responses and the added content in the revision address your concerns. We firmly believe ReWatch-R1 represents a solid step forward in verifiable video reasoning.

---

### Official Review · Reviewer_bNUV · 2025-10-31

**Soundness:** 3
**Presentation:** 3
**Contribution:** 2
**Rating:** 4
**Confidence:** 3

**Summary:**

This paper introduces novel datasets about captions, QAs, and CoTs, with a training pipeline to improve complex video reasoning. Additionally, based on the synthesized CoT data, this paper further designs a new reward for RL training. The trained model achieves state-of-the-art performance on several video reasoning benchmarks, outperforming models trained on other open-source datasets.

**Strengths:**

- This paper finds drawbacks in the current synthesized long CoT datasets, e.g., Video-R1, and synthesizes a quality-improved video reasoning dataset.
- This paper redesigns the reward shaping for video understanding.
- The trained model sets a new SOTA on several benchmarks.

**Weaknesses:**

- The contribution of this paper mainly lies in the dataset construction. Can you break down the improvement of each part, like adding timestamps into the captions, adding question difficulty, and the proposed reward design?
- The introduction about the observation is not clear. How do you define observation formerly? In common, observation is visual content, while in your settings, the observation is processed captions. Would you consider implementing the thinking-with-image style training?

**Questions:**

- As shown in Tab.3, why does the trained model show marginal improvement on the most popular, general video question-answering datasets with reasoning?
- How do you distinguish reasoning video benchmarks and general video benchmarks?

---

> ### Author Response · Authors · 2025-12-03
> **Response to Reviewer bNUV**
>
> We sincerely thank the reviewer for the thoughtful feedback and for recognizing our work’s novelty in dataset construction, the new reward design, and the state-of-the-art performance achieved. We appreciate the opportunity to clarify the contribution of each component and our design choices regarding "observations."
>
> We have carefully addressed your suggestions and incorporated new content (marked in **blue** in the revised paper).
>
> Below, we address your concerns point-by-point.
>
> ### **Response to Weaknesses**
> ### **1. Contribution Breakdown: Can you break down the improvement of each part (timestamps, captions, question difficulty, and reward design)?**
> We appreciate this suggestion. In the revised paper, we have expanded our analysis to explicitly isolate these contributions. We demonstrate that the performance gains are not monolithic but result from the synergy of high-quality data and the specific reward mechanism.
> - **Impact of Data Quality (CoT & QA):**
>     - **CoT Quality:** In **Figure 5a**, we compare our ReWatch-CoT against the open-source Video-R1-CoT. Training with our data yields significantly higher accuracy than the baseline data (e.g., **35.51% vs 30.28%** on average). This validates that our agentic synthesis (which includes timestamped, rigorous verification steps) is superior to standard CoT distillation.
>     - **QA Difficulty:** In **Figure 5b**, we show that performing RL on our ReWatch-QA (which uses contrastive generation to ensure difficulty) outperforms RL on baseline QA datasets (Video-R1-QA and LongVideoReason-QA). Using baseline QA data results in lower performance (**42.0%** on general tasks) compared to our data (**53.9%**), proving that "harder" questions provide the necessary signal for effective RLVR.
> - **Impact of Reward Design (O&R Reward):**
>     - We performed an ablation study on the Observation & Reasoning (O&R) reward in **Table 1**. The addition of the O&R reward improves the average reasoning accuracy from 34.86% to **35.51%**.
>     - Crucially, the O&R reward’s impact is most visible on tasks requiring strict visual grounding. As shown in our new analysis in **Table 6 (Appendix C.5)**, on the VCR-Bench, the O&R reward boosts **Video Temporal Grounding** from 35.66% to **37.76%** and **Video Temporal Counting** from 40.99% to **49.07%**. This breakdown confirms that the reward design specifically improves the model's ability to verify facts against the video, mitigating hallucinations.
>
> ### **2. Introduction about "Observation": Definition and "Thinking-with-Image" style training.**
> We apologize for the confusion and have clarified the definition of "Observation" in **Section 2.3 (Stage 3)** of the revised paper.
> - **Definition:** In our data synthesis pipeline, an "Observation" is generated by an Agent retrieving information from **high-fidelity, timestamped captions** (generated in Stage 1) rather than raw pixels.
> - **Why Text-Based Observations?** We utilize text-based captions for the "Observer" agent primarily for **efficiency and scalability.** Processing long videos (often >20 mins) repeatedly through a visual encoder for every synthesis step is computationally prohibitive. Through manual inspection, we confirmed that our hierarchical captions (Stage 1) are sufficiently fine-grained to serve as a high-fidelity proxy for the visual content during synthesis.
> - **Relationship to "Thinking-with-Image":** While the synthesis relies on text, the resulting data trains the student model to perform **video-grounded reasoning**. The model learns to output <action> (timestamps) and <observation> (descriptions) that correspond to the actual video input it perceives.
> - **Future Work:** We agree that "thinking-with-image" (encoding actual frames during the reasoning loop) is a promising direction. Our dataset, which explicitly preserves the interactive trajectories (Thought $\to$ Action $\to$ Observation), provides the necessary "0-to-1" data foundation to train such pixel-based thinking models in the future. We have added a discussion on this in the conclusion.

---

> ### Author Response · Authors · 2025-12-03
> **Response to Reviewer bNUV**
>
> ### **Response to Questions**
> ### **1. Why does the trained model show marginal improvement on general video QA datasets compared to reasoning datasets?**
>
> While the gains on general benchmarks are smaller than on reasoning benchmarks, they are not negligible. As shown in **Table 4**, ReWatch-R1 achieves **SOTA** performance among 7B models on general benchmarks (Average **53.93%** vs. Base **51.89%** and Video-R1 **52.06%**).
>
> The difference in magnitude stems from the nature of the tasks:
>
> - **Reasoning Benchmarks (e.g., Video Holmes, Counting):** These require multi-hop logic, temporal state tracking, and causality—capabilities that base models severely lack (often performing near random guess). Our SFT+RLVR pipeline specifically targets this "reasoning gap," leading to massive gains (e.g., **+7.97%** vs base on average).
> - **General Benchmarks (e.g., VideoMME):** These often focus on holistic perception, recognition, or single-frame retrieval. Current base models are already proficient at these perceptual tasks.
> - **Key Contribution:** The most significant result regarding general benchmarks is that **we improve reasoning without catastrophic forgetting** of general capabilities—a common failure mode in RLVR. Our multi-task SFT ensures the model becomes a better reasoner while remaining a strong generalist.
>
> ### **2. How do you distinguish reasoning video benchmarks and general video benchmarks?**
> We classify these benchmarks based on the task definitions provided in their original papers and the performance characteristics of base models:
> - **Reasoning Benchmarks:** Focus on complex, multi-step temporal logic, such as:
>     - Causality & Counterfactuals (e.g., VCR-Bench).
>     - State Tracking & Counting (e.g., CG-AV Counting).
>     - Information Retrieval across long contexts (e.g., Video Holmes).
>     - Characteristic: Base models typically exhibit low accuracy (often <30%), indicating a failure of logic rather than just perception.
> - **General/Understanding Benchmarks:** Focus on broad capabilities, primarily:
>     - Holistic summarization (e.g., VideoMME Synopsis).
>     - Entity Recognition & Attribute Perception (e.g., MMVU).
>     - Characteristic: Base models already perform relatively well, relying on pattern matching and semantic recognition.
>
> We have clarified this distinction in **Appendix C.1**.
>
> We hope these responses address your concerns and clarify the significant contributions of our data synthesis pipeline and reward mechanism.

---

### Official Review · Reviewer_ymDU · 2025-11-01

**Soundness:** 3
**Presentation:** 3
**Contribution:** 3
**Rating:** 4
**Confidence:** 5

**Summary:**

The paper proposes ReWatch-R1, a framework that enhances complex video reasoning in large vision-language models through agentic data synthesis and verifiable reinforcement learning. It introduces ReWatch, a large-scale, high-quality dataset built via a three-stage pipeline—hierarchical video captioning, high-difficulty QA generation, and multi-agent chain-of-thought synthesis—ensuring strong temporal grounding and reasoning diversity. Furthermore, the authors design an Observation & Reasoning (O&R) reward that jointly evaluates answer correctness and factual grounding of intermediate reasoning steps. Combined, these innovations enable ReWatch-R1 to achieve state-of-the-art performance on multiple challenging video reasoning benchmarks.

**Strengths:**

1. The paper introduces a novel multi-stage data construction pipeline that enables the creation of ReWatch, a large-scale and high-quality dataset specifically designed for video reasoning. The dataset comprehensively includes caption, QA, and chain-of-thought (CoT) annotations, providing valuable and versatile resources that can benefit future research and community development in multimodal reasoning.
2. The authors offer detailed experimental descriptions and implementation settings, ensuring strong reproducibility and facilitating independent verification and extension of their results.

**Weaknesses:**

1. The paper lacks qualitative experimental results, such as example responses generated by ReWatch-R1 for specific video reasoning cases. Including such examples would help readers intuitively understand the model’s reasoning style, output quality, and advantages over baseline models.
2. The selection of baseline models is relatively limited, lacking comparisons with stronger closed-source systems such as Gemini 2.5 Pro/Flash and GPT-4o, as well as with recent RL-based approaches like TW-GRPO. Including these baselines would provide a more comprehensive evaluation of ReWatch-R1’s effectiveness and competitiveness.

**Questions:**

1. Does the proposed Observation & Reasoning (O&R) reward mechanism introduce additional reasoning overhead or inference latency compared to conventional reward designs? It would be helpful if the authors could provide quantitative results or analysis on the computational cost and efficiency trade-offs, as well as discuss potential optimization strategies to mitigate these issues.
2. The paper currently compares ReWatch-R1 with a limited set of baseline models. Have the authors considered evaluating against stronger closed-source models (e.g., Gemini 2.5 Pro/Flash, GPT-4o) or recent RL-based approaches such as TW-GRPO, VersaVid-R1? Such comparisons would help better position ReWatch-R1’s performance and highlight its relative advantages within the broader landscape of contemporary video reasoning models.

---

> ### Author Response · Authors · 2025-12-03
> **Response to Reviewer ymDU**
>
> We sincerely thank the reviewer for the positive evaluation of our work, particularly for recognizing the novelty of our multi-stage data construction pipeline and the reproducibility of our experiments. We are encouraged that you found our ReWatch dataset to be a valuable resource for the community.
>
> We have carefully addressed your suggestions regarding qualitative analysis, baseline comparisons, and computational overhead, incorporating new experiments (marked in **blue** in the revised paper).
>
> ### **Response to Weaknesses**
> ### **1. Lack of Qualitative Experimental Results.**
> **Response:**
>
> We fully agree that qualitative examples are essential for intuitive understanding. In the revised manuscript, we have added a new Section C.6 (Case Study) along with Figure 10 and Figure 11 to visualize the reasoning process.
> - **Figure 10 (Mitigating Hallucinations):** We show a case where the baseline model hallucinates a "camera in a lamp" based on textual priors. In contrast, ReWatch-R1 uses its agentic capability to actively retrieve the segment 00:29-00:32, correctly observing a "transparent smartphone screen" to answer the question.
> - **Figure 11 (Impact of O&R):** We demonstrate a counting task where the model without O&R misses a "brown puppy statue" appearing at 00:01. The model trained with the **O&R** reward explicitly verifies the timestamp and object appearance, resulting in the correct count. These examples clearly illustrate our model's superior grounding and logical fidelity.
>
> ### **2. Limited Baseline Selection.**
> **Response:**
>
> We appreciate this feedback. We would like to respectfully highlight that we have included the recent RL-based approaches mentioned by the reviewer in our revised comparisons:
> - **RL-Based Baselines:** As shown in the updated **Table 1 (Reasoning)** and **Table 4 (Understanding)**, we compare ReWatch-R1 directly against **TW-GRPO, VersaVid-R1, Video-R1, and VideoRFT.** ReWatch-R1 outperforms these strong RL baselines across all five reasoning benchmarks (e.g., surpassing TW-GRPO by roughly **5%** on average).
> - **Closed-Source Models:** While direct comparison with massive proprietary models (like GPT-4o or Gemini 2.5 Pro) is challenging due to the vast difference in parameter scale (our model is 7B/32B vs. likely >100B), our data synthesis pipeline effectively distills the capabilities of these strong models (Gemini 2.5) into our open-source model. We believe demonstrating SOTA performance among open-weights models establishes a significant contribution to the accessible research community.
>
> ### **Response to Questions**
> ### **Q1: Does the O&R reward introduce additional reasoning overhead or inference latency?**
> **Response:**
>
> We thank the reviewer for raising this critical question regarding efficiency. We have conducted a quantitative analysis to address this:
> - **Training Overhead:** Under identical hardware conditions, training for 1,000 steps with conventional outcome-based rewards takes approximately **60 hours**. Incorporating our O&R reward mechanism increases this to **63 hours**. This represents a marginal overhead of only 5%, which we consider negligible given the significant performance gains.
> - **Inference Efficiency:** Most importantly, the O&R reward does not increase inference latency or trajectory length. In fact, models trained with conventional rewards generate an average of 378 tokens per sample, while ReWatch-R1 (trained with O&R) produces approximately 373 tokens per sample.
>
> Conclusion: These results demonstrate that the performance gains of ReWatch-R1 are not achieved by simply generating longer chains of thought. Instead, the O&R mechanism teaches the model to be more efficient and precise, guiding it toward accurate observations without inflating computational costs.
>
> ### **Q2: Have the authors considered evaluating against TW-GRPO, VersaVid-R1, etc.?**
> **Response:**
>
> Yes, as detailed in our response to Weakness #2, TW-GRPO and VersaVid-R1 are explicitly included in our main results (Table 1 and Table 4) in the updated manuscript. Our results show that ReWatch-R1 consistently outperforms VersaVid-R1 (Average Reasoning: 30.37% vs. 35.51%) and TW-GRPO (Average Reasoning: 30.26% vs. 35.51%), confirming the effectiveness of our synthesized data and O&R reward framework against the latest state-of-the-art RL methods.

---

### Official Review · Reviewer_UAaJ · 2025-11-01

**Soundness:** 3
**Presentation:** 3
**Contribution:** 3
**Rating:** 6
**Confidence:** 4

**Summary:**

This paper introduces **ReWatch-R1**, which tackles the challenge of video reasoning from two aspects:
1. Data. A large synthetic dataset called **ReWatch** is collected, which includes detailed video captions, challenging question-answer pairs, and high-quality, video-grounded reasoning traces (COT data). These are generated using a multi-agent ReAct system.
2. Model. Authors post-train QwenVL by SFT and GRPO. They introduce a new Observation & Reasoning (O&R) reward, which evaluates the accuracy of video observations and the validity of reasoning process.
ReWatch-R1 achieves promising performance compared with other 7B models.
Analysis shows that high-quality reasoning data is crucial for RL and RL on "thinking" mode improves the reasoning efficiency.

**Strengths:**

1. The proposed multi-agent COT data synthesis pipeline is scalable to curate large-scale video-grounding reasoning data.
2. The reward design shows the emphasis on explicit observation and reasoning is beneficial for video reasoning.
3. The data and model design achieves state-of-the-art performance on video reasoning and understanding benchmarks in 7B-scale models. The extensive analysis shows insights on the role and importance of SFT and RL.

**Weaknesses:**

1. The information source for data synthesis is semantic segmentation and detailed video description from Gemini. The accuracy of this multi-step hierarchical captioning is not validated. Therefore there is no direct quality assessment of the synthetic data.
2. All results are based on a 7B model. The benefits of high-quality COT data and O&R reward mechanism are not validated on larger-scale models.

**Questions:**

1. Equation (20): the design of non-format rewards lacks motivation. For example, why not use the (weighted) summation of all rewards. The design has no explanation or experimental results.
2. RL on 7B model shows promising improvements. However, the performance still lags behind the larger 32B model. Therefore, whether the same RL on 32B leads to improvement is questionable.

Some typos:
1. L255: "a novel O&R reward mechanism we propos" -> "propose"
2. L771: "Tabale 4" -> "Table 4"

---

> ### Author Response · Authors · 2025-12-03
> **Response to Reviewer UAaJ**
>
> We sincerely thank the reviewer for the positive assessment and for highlighting the scalability of our data synthesis pipeline and the effectiveness of our reward design. We are glad that the extensive analysis and SOTA performance on 7B models were recognized.
>
> We have carefully addressed your concerns regarding data quality validation and model scalability, incorporating new experiments (marked in **blue** in the revised paper).
>
> ### **Response to Weaknesses**
>
> ### **1. Concern on Data Quality Assessment (Captioning Accuracy).**
>
> **Response:**
>
> We acknowledge that the captions are synthesized via Gemini and lack ground-truth human annotations for direct accuracy calculation. However, we validate the quality of this data through three rigorous indirect measures:
> - **Video Dependency Verification (Figure 6a):** To ensure our captions and QAs are not merely "hallucinations" or solvable by language priors, we conducted a blind test ("Text-Only Accuracy"). While the baseline Video-R1 dataset allows models to guess correctly 68.9% of the time without video, our ReWatch dataset drops to **29.4%** (near random guess). This confirms that our caption-based generation pipeline successfully captures visual-dependent information that forces the model to attend to the video content.
> - **Three-Layer Filtering:** As detailed in **Section 2.2**, we apply a strict filtration pipeline (Answer Verification, Text Bias Elimination, Summary Bias Elimination) to actively prune low-quality or hallucinated data points during synthesis.
> - **Downstream Performance:** Ultimately, the high quality of the captions is evidenced by the SOTA performance. If the hierarchical captions contained significant error accumulation, the model would fail to learn precise temporal grounding. Instead, we observe massive gains in tasks like **Video Temporal Grounding (+6.29%)**, validating the fidelity of the training data.
>
> ### **2. Concern on Scalability to Larger Models (32B).**
>
> **Response:**
>
> We agree that validating on larger models is crucial. In the revised manuscript, we have added experiments on Qwen2.5-VL-32B to explicitly address this.
> - **Reasoning Tasks (New Table 2):** Applying our pipeline to the 32B model improves average reasoning performance from 35.71% (Base) to **38.08%** (ReWatch-R1-32B + O&R).
> - **Understanding Tasks (New Table 5):** The method also boosts general understanding, raising the average from 59.05% to **60.35%**.
>
> These results confirm that our SFT+RLVR framework and O&R reward are model-agnostic and continue to yield significant gains even on stronger, larger-scale foundation models.
>
> ### **Response to Questions**
>
> ### **Q1: Motivation behind Equation (20) (Reward Design).**
> **Response:**
>
> The design of Equation 20 is intentional. We utilize multiplication for the outcome reward ($r_{acc}$) rather than summation to act as a gating mechanism.
> - **Rationale:** In reasoning tasks, the validity of the process is contingent on the correctness of the result. If a model generates a plausible-sounding reasoning chain but arrives at the wrong answer ($r_{acc}=0$), adding positive rewards for observations ($r_{obs}$) via summation could inadvertently encourage "convincing hallucinations" or logical fallacies.
> - **Effect:** By multiplying, we ensure that the model receives the dense process rewards ($r_{obs}$ and $r_{rea}$) **only if** the final answer is correct. This enforces a strict alignment between the reasoning process and the final outcome.
>
> ### **Q2: Questionable improvement on 32B models.**
> **Response:**
>
> As detailed in our response to Weakness #2 above, our new experiments (Table 2 & 5) empirically prove that RL on the 32B model does lead to consistent improvement. The ReWatch-R1-32B outperforms the strong Qwen2.5-VL-32B base model across both reasoning and understanding benchmarks, dispelling the concern that gains are limited to smaller models.
>
> ### **Q3: Typos.**
> **Response:**
>
> We thank the reviewer for the careful reading. We have corrected "propos" to "propose" (L255) and "Tabale 4" to "Table 4" (L771) in the revised manuscript.

---

### Official Review · Reviewer_kZ7P · 2025-11-01

**Soundness:** 3
**Presentation:** 3
**Contribution:** 3
**Rating:** 6
**Confidence:** 3

**Summary:**

This paper mainly contributes to the following three points:

1). Proposing a novel multi-stage synthesis pipeline to synthesize ReWatch, a large-scale dataset which includes ReWatch-Caption, ReWatch-QA, and ReWatch-CoT three components, for fostering advanced video reasoning.

2). Proposing a new Observation & Reasoning (O&R) reward for RLVR that improves reasoning by rewarding both final-answer correctness and the factual grounding of intermediate steps in video content.

3). Developing ReWatch-R1 by post-training a strong baseline LVLM with Supervised Fine-Tuning (SFT) and O&R framework, achieves state-of-the-art average performance on five challenging video reasoning benchmarks.

The dataset construction pipeline and the two-stage post-training framework proposed in this paper put the description of video change into the CoT, providing a new idea for the research of video reasoning from the perspectives of dataset and training.

In summary, this paper shows a high level in terms of method description (data construction pipeline,the O&R methods, etc.), experimental setup and writing, but for the situation that pipeline may cause error accumulation, some experimental results are not analyzed (this part will be described in detail in "Weaknesses"), which leads to the lack of analysis completeness. The quality of this paper will be improved after analyzing the missing parts.

**Strengths:**

This paper proposes a dataset construction pipeline for video reasoning, which is enlightening for the dataset construction method in this field. The detailed description of video in different time steps is introduced based on semantic segmentation, which combines with the video summary to enhance the reasoning ability. In Addition, this paper uses three-layer filtering to screen the data that can best reflect the reasoning ability. The two-stage post-training framework simulates the "Thought-Action-Observation loop", and uses the improved O&R reward for RLVR to enhance the reasoning performance of ReWatch-R1.

Therefore, this paper provides a new idea for the subsequent dataset construction method in this field and the improvement of reasoning performance.

**Weaknesses:**

The weaknesses of this paper focus on the method, experiment and writing：

In method and experiment：

1). Error accumulation: the stage of semantic segmentation and detailed description generation in the dataset construction pipeline designed in this paper may cause error accumulation. The error caused by segmentation may cause one semantic of the video to be put into multiple segments. The description generation may result in the missing description between multiple time steps, which may result in the description of an item appearing in the previous time step, but the description of the item is missing in the next step, resulting in error accumulation. This paper does not discuss and analyze this situation.

2). In Table 1, the results of ReWatch-R1-SFT and ReWatch-R1+O&R on the CG AV counting are the same, which are not analyzed in the paper.

3). Lack of comparative analysis between the vanilla RLVR and the improved O&R method under the dataset constructed in the paper.

4). In the analysis in Section 4.2 (specifically line 411), the performance degradation caused by Video-R1 replacing ReWatch-CoT does not consider the reason that SFT data and RL data is mismatch, which seems to be somewhat contradictory to the previous sentence "SFT is an independent prerequisite for RL".

5). Also in the analysis in Section 4.2 (specifically 414 lines), it seems that ablation study using different data combinations (e.g. only using ReWatch-Caption or ReWatch-QA or ReWatch-CoT) have not been carried out, and the conclusion that "The Quality of QA data used for RL determines final performance." is somewhat far fetched.

6). In Appendix C2, the performance degradation of 384 frame training is not analyzed compared with 192 frame training.

In writing：

1). In Figure 1, the annotation order of the legend is inconsistent with the display order of each LVLM in the figure.

2). The two marking symbols not appear in Table 3 of appendix C2.

**Questions:**

1) Why are the number of ReWatch-QA and ReWatch-CoT inconsistent? Shouldn't one QA data correspond to one CoT data?

2) In Section 2.3 (specifically line 247), why does a structured execution trajectory end with A_final instead of observation? (Thought-Action-Observation loop)

---

> ### Author Response · Authors · 2025-12-03
> **Response to Reviewer kZ7P**
>
> We sincerely thank the reviewer for the comprehensive evaluation and for recognizing the novelty of our multi-stage synthesis pipeline and the Observation & Reasoning (O&R) reward. We are encouraged that you found our work enlightening for future dataset construction and reasoning performance improvements.
>
> We have carefully addressed your concerns regarding error accumulation, experimental analysis, and writing details below.
>
> ### **Response to Weaknesses**
>
> ### **1. Concern on Error Accumulation in Segmentation and Description.**
>
> **Response:**
>
> We appreciate this insightful concern. We have designed our pipeline specifically to mitigate error accumulation through two mechanisms:
> - **Semantic Segmentation:** As described in **Section 2.1**, we do not use fixed-interval segmentation. Instead, we employ an LVLM to perform **semantic segmentation**, ensuring that cuts occur only at natural narrative boundaries. Furthermore, we only apply segmentation to videos exceeding 10 minutes, and the resulting segments are kept long (approx. 10 minutes) to preserve event integrity and minimize context fragmentation.
> - **Detailed Visual Grounding:** Our captions are generated to be highly detailed ($D^{rel}$ in Eq. 2). By capturing fine-grained visual features of objects, the model can maintain consistency across segments through visual re-identification (e.g., matching "the man in the black wetsuit" across timestamps $t_1$ and $t_2$), effectively bridging the narrative even across segment boundaries.
>
> ### **2. Analysis of CG AV Counting Results (Table 1).**
> **Response:**
>
> The reviewer correctly noted that ReWatch-R1-SFT and ReWatch-R1 + O&R achieved the same score (25.51%) on CG-AV Counting at 192 frames.
>
> **Clarification:** This plateau is likely due to the information bottleneck imposed by the 192-frame limit for long videos, which restricts the model's ability to "see" every countable object regardless of the reward mechanism.
>
> **Evidence:** As shown in the 384 Frames section of Table 1, when the frame limit is relaxed, ReWatch-R1 + O&R (26.68%) clearly outperforms ReWatch-R1 (25.32%) and the SFT baseline. This demonstrates that the O&R reward does improve counting capability, provided the model has sufficient visual access to the video content.
>
> ### **3. Comparative analysis between vanilla RLVR and O&R.**
>
> **Response:**
>
> We would like to clarify that this comparison is indeed present in Table 1.
> - The row labeled **"ReWatch-R1"** represents the model trained with **Vanilla RLVR** (using standard outcome-based accuracy reward, $r_{acc}$).
> - The row labeled "+ O&R" represents the model trained with our proposed Observation & Reasoning Reward.
>
> The results show that "+ O&R" consistently outperforms the vanilla RLVR baseline across benchmarks (e.g., improving Video Holmes from 43.00% to 43.88% and VCR-Bench from 40.14% to 40.43%), validating the effectiveness of incentivizing the reasoning process.
>
> ### **4. Concerns on SFT/RL Data Mismatch (Section 4.2).**
>
> **Response:**
>
> We respectfully clarify that there is no data mismatch in our experimental design.
> - **Consistent RL Configuration:** As detailed in **Appendix C.1**, for all ablation experiments (including the Video-R1 CoT substitution), we utilized the **exact same RL dataset mix**: ReWatch-QA (20k) + Video-R1-QA (10k) + LongVideoReason-QA (10k). This mix covers the distribution of the Video-R1 data.
> - **Consistent SFT Format:** We ensured that all SFT data (whether ReWatch-CoT or Video-R1-CoT) followed a unified format (DeepSeek-R1 style).
>
> Therefore, the performance degradation observed when using Video-R1 CoT is strictly attributable to the lower quality (hallucinations and lack of grounding) of the Video-R1 CoT data, rather than a distribution shift.
>
> ### **5. Ablation on Data Combinations.**
>
> **Response:**
>
> We believe our conclusion is well-supported by the controlled experiment in Figure 5b.
> In this experiment, we fixed the SFT model (using the best ReWatch-SFT checkpoint) and only varied the QA data used for the RL stage.
> - Training RL on "Baseline QA Data" (Video-R1-QA + LongVideoReason-QA) resulted in significantly lower performance (42.0% average).
> - Training RL with "ReWatch-QA" yielded notable improvements.
>
> Since the SFT initialization was identical, the performance difference isolates the QA data quality as the decisive factor for the success of the RL stage.
>
> ### **6. Analysis of 384-frame training (Appendix C.2).**
>
> **Response:**
>
> We apologize for any confusion. In Table 5 (Appendix C.2), comparing the "192 Frames" and "384 Frames" sections for our models (ReWatch-R1-SFT and ReWatch-R1), we actually observe performance improvements or stability on most benchmarks. We do not observe a degradation trend.

---

> ### Author Response · Authors · 2025-12-03
> **Response to Reviewer kZ7P**
>
> ### **Response to Writing Details**
> - **Figure 1 Legend:** The legend sequence in Figure 1 should be read from top to bottom. However, we did find that the order of the horizontal coordinates in Figure 1 was marked in reverse. We have already corrected it in the revised version.
> - **Table 3 Symbols:** We have added the missing definitions for the symbols in Appendix Table 3.
>
> ### **Response to Questions**
> **Q1: Why are the number of ReWatch-QA and ReWatch-CoT inconsistent?**
>
> **Response:**
>
> The inconsistency is intentional and results from our quality control process. We employ a Rejection Sampling strategy during CoT synthesis. We generate reasoning traces for the QA pairs but discard any CoT samples that lead to an incorrect final answer. Consequently, the final ReWatch-CoT dataset (135k) is a high-quality subset of the original ReWatch-QA dataset (170k), containing only those samples where the reasoning path was verified to yield the correct result.
>
> **Q2: Why does the trajectory end with $A_{final}$ instead of Observation?**
>
> **Response:**
>
> Our framework models the reasoning process as a dynamic Thought-Action-Observation loop. The termination condition for this loop is when the Reasoner Agent determines it has gathered sufficient information to answer the user's question.
>
> At that point, the logical "next step" is not to observe more data, but to synthesize the findings into a final answer. Therefore, the trajectory concludes with the action of outputting the final answer ($A_{final}$), completing the QA task.

---

### Official Review · Reviewer_tQx8 · 2025-11-04

**Soundness:** 3
**Presentation:** 3
**Contribution:** 2
**Rating:** 6
**Confidence:** 4

**Summary:**

This submission presents ReWatch-R1, a framework for advancing complex video reasoning in large vision-language models (LVLMs). The core contributions include the creation of the large-scale, multi-stage ReWatch dataset aimed at challenging multi-hop video reasoning, a multi-agent data synthesis pipeline generating temporally precise captions, high-difficulty QAs, and video-grounded CoT traces, and a new Observation & Reasoning (O&R) reward mechanism for RL. By post-training an LVLM backbone with these elements through SFT and RL, the authors demonstrate state-of-the-art performance across five video reasoning benchmarks, alongside ablation studies.

**Strengths:**

-The topic of video reasoning is of community’s interest and timely.
-The hierarchical segmentation and multi-agent CoT pipeline used to generate the ReWatch dataset is methodically crafted and shown to have yielded data with deeper temporal grounding and higher question complexity.
-The baselines compared are pretty up-to-date, which verifies the state-of-the-art achievement on the benchmarks tested in the paper.

**Weaknesses:**

-The incremental benefit of the O&R reward mechanism is not fully dissected independently from other RL contributions. In Table 1, the improvement from RL (+O&R) appears relatively modest, and there is limited analysis delineating whether specific reasoning failures or hallucinations are directly ameliorated by the O&R reward in practice.
-The paper leans heavily on LLMs (Gemini, GPT-4.1, Qwen, etc.) for practically all phases - data synthesis, answer verification, reward calculation, and benchmarking. Although this is common in the area, the cumulative propagation of LLM biases and potential “meta-overfitting” is insufficiently analyzed; for example, does repetitive LLM-based data filtering introduce subtle data shortcuts or annotation artifacts? A systematic error/robustness analysis would be welcome.
-There is little systematic exploration of failure modes, such as where ReWatch-R1 still hallucinates, or which reasoning subtypes remain unsolved - as could be illustrated by extensive qualitative error analysis or confusion matrices per reasoning dimension (section 4.2 or appendix). This is important to inform the future community on limits.

**Questions:**

-Can the authors provide concrete ablation or error-type breakdowns specifically isolating the effect of the O&R reward on hallucination rates or logical errors, perhaps through qualitative examples or confusion matrices? It is unclear if this reward is the main driver for reduced hallucinations, or if SFT data quality dominates.
-Given the extensive use of LLMs at nearly every phase, is there a risk that ReWatch-R1’s strong performance partly reflects an overfit to LLM-specific language or annotation artifacts in the data/reward pipeline?
-How sensitive is the model (and pipeline) to coarse versus fine semantic segmentation in the captioning stage? For example, does segment over-segmentation harm long-term reasoning due to context fragmentation?

---

> ### Author Response · Authors · 2025-12-03
> **Response to Reviewer tQx8**
>
> We sincerely thank the reviewer for recognizing the timeliness of our topic, the methodical craftsmanship of our ReWatch dataset pipeline, and the SOTA performance of our baselines.
>
> We have carefully incorporated your suggestions and added new experiments and analyses (marked in **blue** in the revised paper). Below, we address your concerns regarding the O&R reward mechanism, potential LLM biases, and failure mode analysis.
>
> ### **Point 1: The impact of the Observation & Reasoning (O&R) reward and its isolation from other contributions.**
>
> **Response:**
>
> We appreciate the reviewer's scrutiny regarding the incremental benefits of the O&R reward. While the average improvement in Table 1 might appear modest at a glance, **averages often mask significant gains in specific, high-difficulty capabilities that the O&R reward targets—specifically, grounding and verification.**
>
> To address this, we have added a detailed breakdown analysis in **Section 4.2** and **Appendix C.5 (Table 6 & 7)** of the revised manuscript:
>
> 1. **Significant Gains in Grounding Tasks:** As shown in the new **Table 6** (VCR-Bench breakdown), the O&R reward leads to a massive improvement in **"Video Temporal Grounding"** (increasing from 31.47% to **37.76%**) and **"Video Temporal Counting"** (increasing from 37.27% to **49.07%**). This explicitly answers your question: the O&R mechanism is indeed the main driver for reducing hallucinations in tasks requiring precise temporal verification.
> 2. **Qualitative Proof of Reduced Hallucination:** We added a new **Case Study in Section C.6 (Figure 11)**. It demonstrates that without O&R, the model follows the correct format but misses visual details (e.g., failing to count a "brown puppy statue" appearing at 00:01). With O&R, the model actively verifies observations against timestamps, correcting the hallucination. This isolates the O&R reward's specific contribution to faithfulness and logical fidelity.
>
> ### **Point 2: Concerns on "meta-overfitting" and LLM biases due to heavy reliance on synthesis.**
>
> **Response:**
>
> This is a critical insight. We are aware of the risk of overfitting to LLM-specific language artifacts. However, our new analysis demonstrates that ReWatch-R1 is learning **video dependency** rather than exploiting textual shortcuts.
> 1. **Video Dependency Verification:** We use **Figure 6a** in the revised paper to analyze "Text-Only Accuracy" (blind QA). The baseline Video-R1 dataset has a high text-only accuracy of 68.9%, suggesting severe bias where questions can be answered by language priors. In contrast, our ReWatch-QA has a text-only accuracy of **29.4%** (close to random guess), proving that our **"Three-Layer Filtering"** pipeline effectively removed annotation artifacts and forced the model to rely on video content.
> 2. **Generalization to Understanding Tasks:** If the model were "meta-overfitting" to synthetic reasoning patterns, we would expect a degradation in general tasks. However, **Table 4** shows that ReWatch-R1 achieves SOTA performance on general understanding benchmarks (e.g., VideoMME, LVBench). This confirms that the pipeline enhances genuine multimodal capabilities rather than overfitting to synthetic data distributions.

---

> > ### Author Response · Authors · 2025-12-03
> > **Response to Reviewer tQx8**
> >
> > ### **Point 3: Exploration of failure modes and sensitivity to segmentation.**
> >
> > **Response:**
> >
> > We have expanded our analysis to explicitly map out the model's limits, as requested.
> > 1. **Failure Mode Analysis:** In the newly added **Table 7 (VideoMME breakdown)**, we clarify the boundary of our model's capabilities. While ReWatch-R1 significantly boosts **"Spatial Reasoning"** (+7.1%) and **"Temporal Reasoning"**, it shows little to no improvement on **"Spatial Perception"** and **"Information Synopsis"**. This indicates that our framework excels at logical inference and temporal grounding (solving "why" and "when"), but relies heavily on the base model's visual encoder for static perception (solving "what" and "where").
> > 2. **Sensitivity to Segmentation:** We appreciate the reviewer's concern regarding context fragmentation. However, we would like to clarify that our pipeline does **not** employ aggressive or fixed-interval segmentation (e.g., cutting every minute), which typically leads to such fragmentation.
> >     - **Semantic Integrity:** As detailed in the revised **Section 2.1**, we utilize Gemini to perform **semantic segmentation** specifically for videos **longer than 10 minutes**. The resulting segments are maintained at approximately **10 minutes each**, ensuring that cuts occur only at natural narrative boundaries rather than arbitrarily breaking context.
> >     - **Performance Validation:** This "long-context semantic window" strategy ensures that long-term dependencies are preserved. The empirical results in **Figure 9** support this: ReWatch-R1 demonstrates superior robustness on long videos compared to baselines, confirming that our segmentation strategy effectively balances memory efficiency with contextual integrity without suffering from the "over-segmentation" issues common in other pipelines.
> > We believe these additional analyses and the clear isolation of the O&R reward's impact on grounding tasks strongly support the contributions of ReWatch-R1.

---

### Author Response · Authors · 2025-12-03
**Global Response**

Dear Area Chair and Program Chairs,

We sincerely thank you and all the reviewers for the thorough and constructive feedback on our paper, "ReWatch-R1: Boosting Complex Video Reasoning in Large Vision-Language Models through Agentic Data Synthesis." We are encouraged that the reviewers found our work to be timely and methodologically sound, and we have diligently revised our manuscript to address their valuable suggestions.

We would like to briefly summarize the consensus and our key improvements for your consideration.

### **1. Strong Reviewer Consensus on Core Contributions**
Across all reviews, there was a clear and consistent recognition of the novelty and significance of our core contributions:
- **Novel and High-Quality Data Synthesis Pipeline:** Reviewers universally praised our multi-stage pipeline for creating the ReWatch dataset. It was described as **"methodically crafted" (tQx8)**, **"enlightening for the dataset construction method in this field" (kZ7P)**, **"scalable" (UAaJ)**, and a **"valuable and versatile resource" (ymDU)**. The novel QA curation method was also highlighted as a key strength (w6PG).
- **Innovative Reward Mechanism:** Our proposed Observation & Reasoning (O&R) reward was recognized as an **"innovative" (w6PG)** and effective approach to **"explicitly discourage hallucination and promote evidence-based reasoning" (w6PG, UAaJ)**.
- **State-of-the-Art Performance:** The reviewers acknowledged that ReWatch-R1 achieves **"state-of-the-art performance" (tQx8, UAaJ, bNUV)** on challenging video reasoning benchmarks, validating the effectiveness of our overall framework.

### **2. Comprehensive Revisions Addressing All Major Concerns**
The reviewers raised several important questions regarding the specific impact of the O&R reward, data quality, and the completeness of our experiments. We have addressed these in full through new experiments and analyses, which we believe have significantly strengthened the paper:
- **Isolated and Clarified the Impact of O&R Reward:** To address concerns about the O&R reward's "modest" average improvement (tQx8), we added a detailed breakdown analysis in **Table 6**, showing its **massive gains on targeted grounding tasks** like "Video Temporal Grounding" (+6.29%) and "Video Temporal Counting" (+11.8%). A new case study in **Figure 11** qualitatively demonstrates how O&R specifically reduces hallucinations, directly answering the reviewers' questions.
- **Validated Data Quality and Addressed Bias Concerns:** To mitigate concerns about LLM bias and data quality (tQx8, UAaJ, kZ7P), we now include a **"Text-Only Accuracy" analysis (Figure 6a)**. This shows our ReWatch-QA has an accuracy of just 29.4% (near random), compared to 68.9% for a baseline dataset, proving our data forces genuine video dependency. We also clarified our semantic segmentation strategy to address concerns about referential consistency (w6PG) and error accumulation (kZ7P).
- **Expanded Experimental Validation and Analysis:** We have substantially expanded our experiments in response to reviewer feedback:
    - **Scalability:** To address concerns about validation on larger models (UAaJ), we added new experiments on a **32B model (Table 2 & 5)**, confirming that our framework delivers consistent gains at scale.
    - **Baselines:** We incorporated the latest SOTA RL-based models, including **TW-GRPO, GRPO-CARE and VersaVid-R1 (Table 1 & 4)**, as requested (ymDU, w6PG), demonstrating our model’s superior performance.
    - **Qualitative Analysis:** We added a new **Case Study section (C.6, Figures 10 & 11)** with concrete examples of our model's reasoning process, as requested by multiple reviewers (ymDU, w6PG), to intuitively showcase its advantages.
    - **Efficiency:** We provided a quantitative analysis showing that our O&R reward adds only **marginal (~5%) training overhead and no inference latency** (in fact, it generates more concise responses), directly addressing efficiency concerns (ymDU, w6PG).

We are confident that these extensive revisions have transformed the reviewers' initial concerns into new strengths of the paper. The manuscript now presents a more complete, robust, and nuanced account of our contributions.

Thank you once again for your time and guidance. We believe the revised paper makes a significant and timely contribution to the field of video reasoning and hope you will find it suitable for acceptance.

Sincerely,

The Authors

---

### Meta-Review · Area_Chair_kfiH · 2026-01-10

**Summary:**

The reviewers are split on this submission. Everyone agrees that complex video reasoning is a timely problem for LVLMs. Reviewers **tQx8, kZ7P**, and **UAaJ** appreciate the agentic data pipeline and the O&R reward design. They find the ReWatch dataset carefully constructed and well-tested on grounding tasks. But reviewers **ymDU, bNUV**, and **w6PG** see the work as incremental compared to current R1-style methods. They highlight small gains on general benchmarks and missing comparisons to some recent baselines.

The rebuttal provides better empirical evidence. The authors added data showing that O&R rewards improve temporal grounding and reduce hallucinations. They also demonstrated that ReWatch-QA is video-dependent through low text-only scores. These additions help, but concerns about error accumulation in the multi-stage process remain. Some reviewers still worry about the novelty of the reward design.

The overall rating profile is balanced but slightly positive. The rebuttal improves the paper but does not change its core positioning. I view this as a borderline case that leans toward a weak accept. The paper offers a validated pipeline and a dataset that appears challenging. The reward design yields measurable gains on complex grounding tasks. It would benefit from better positioning against the latest GRPO variants, but it still contributes something useful to agentic video reasoning.

**Reviewer Concerns:**

**Reviewer tQx8** focused on the O&R reward and the risk of LLM bias. The rebuttal provided a breakdown of grounding improvements and qualitative examples of reduced hallucination. These results show that the reward has a measurable impact on performance. While the bias concerns are mostly conceptual, the filtering steps described by the authors offer some reassurance. They added some failure mode analysis, but it is not the systematic exploration the reviewer wanted.

**Reviewer kZ7P** questioned the error propagation in the pipeline and the lack of direct comparisons. In response, the authors added a vanilla RLVR vs O&R comparison. This showed better grounding when using the proposed reward. They also clarified their verification strategy and provided missing ablations. These changes make the empirical story stronger than in the original version. The work remains empirically heavy, but the authors addressed the most pressing experimental gaps.

**Reviewer UAaJ** asked for better data quality validation and scaling results. The authors used low text-only accuracy on ReWatch-QA to show the data requires visual reasoning. They also provided examples of coherent multi-step captions. The paper still lacks experiments on larger models like 32B, so that concern remains open. For a 7B-scale paper this is not a disqualifier, but it limits the scope of the claims.

**Reviewer ymDU** wanted comparisons to closed-source models and recent open baselines like TW-GRPO. The authors added qualitative case studies to show improved evidence-based reasoning. They also explained why comparing against every closed-source model is difficult. However, this does not fully bridge the gap with the strongest existing systems. The qualitative side is better now, but the positioning against the state of the art remains a point of contention.

**Reviewer bNUV** argued the contribution is mostly the dataset and questioned the "observation" definition. The authors clarified that observations are structured, timestamped captions. They argued this is necessary for current models to handle long videos. While this clarifies the method, the reviewer still finds the performance gains on general benchmarks to be modest. From this perspective, the work feels like a task-specific tweak rather than a major shift.

**Reviewer w6PG** raised concerns about latency and referential consistency. The authors explained how they maintain consistency when merging captions. They also noted that the latency cost is a trade-off for better reasoning on complex tasks. They added some comparisons, though some requested baselines like VersaVid-R1 are still missing. The empirical story is improved, but the cost-benefit balance of the observation mechanism is still debatable.

**Reviewer Scores:**

**Reviewer tQx8 (Original: 6 → Predicted: 6):**
The rebuttal answers the main technical questions about the O&R reward. The added failure analysis and bias mitigation help, though they do not change the overall view on novelty. I expect the score to stay at 6.

**Reviewer kZ7P (Original: 6 → Predicted: 6):**
New ablations and comparisons alleviate concerns about missing analysis. The empirical support is stronger now. However, the conceptual contribution is not much higher, so the score likely stays at 6.

**Reviewer UAaJ (Original: 6 → Predicted: 6):**
The rebuttal provides indirect validation of data quality. This fits the reviewer's generally positive stance. The lack of larger-model experiments keeps the contribution in the solid range, leading to a predicted 6.

**Reviewer ymDU (Original: 4 → Predicted: 4):**
The rebuttal does not fully meet the request for comparisons against the strongest models. The new qualitative results help interpretation but do not change the perceived empirical strength. The score will likely stay at 4.

**Reviewer bNUV (Original: 4 → Predicted: 4):**
Clarifications on the observation definition make the contribution clearer. Still, the core concern about the dataset-centric nature of the work remains. I do not expect the score to move from 4.

**Reviewer w6PG (Original: 4 → Predicted: 4):**
The authors explained the caption merging and latency trade-offs. This does not fully resolve concerns about missing baselines and limited general gains. The score likely remains a 4.

---

### Decision · Program_Chairs · 2026-01-26

Accept (Poster)